# Large-scale and Fine-grained Vision-language Pre-training for Enhanced CT Image Understanding

**Zhongyi Shui**[1,3,4]    **Jianpeng Zhang**[1,3,5] *    **Weiwei Cao**[1,3,5]    **Sinuo Wang**[1]    **Ruizhe Guo**[3,4]
**Le Lu**[1]    **Lin Yang**[4]    **Xianghua Ye**[2]    **Tingbo Liang**[2]    **Qi Zhang**[2]    **Ling Zhang**[1]

[1]DAMO Academy, Alibaba Group

[2]The First Affiliated Hospital of College of Medicine, Zhejiang University, China
[3]Zhejiang University, China    [4]Westlake University, China    [5]Hupan Lab, 310023, China
jianpeng.zhang0@gmail.com

## ABSTRACT

Artificial intelligence (AI) shows great potential in assisting radiologists to improve the efficiency and accuracy of medical image interpretation and diagnosis. However, a versatile AI model requires large-scale data and comprehensive annotations, which are often impractical in medical settings. Recent studies leverage radiology reports as a naturally high-quality supervision for medical images, using contrastive language-image pre-training (CLIP) to develop language-informed models for radiological image interpretation. Nonetheless, these approaches typically contrast entire images with reports, neglecting the local associations between imaging regions and report sentences, which may undermine model performance and interoperability. In this paper, we propose a fine-grained vision-language model (fVLM) for anatomy-level CT image interpretation. Specifically, we explicitly match anatomical regions of CT images with corresponding descriptions in radiology reports and perform contrastive pre-training for each anatomy individually. Fine-grained alignment, however, faces considerable false-negative challenges, mainly from the abundance of anatomy-level healthy samples and similarly diseased abnormalities, leading to ambiguous patient-level pairings. To tackle this issue, we propose identifying false negatives of both normal and abnormal samples and calibrating contrastive learning from patient-level to disease-aware pairing. We curated the largest CT dataset to date, comprising imaging and report data from 69,086 patients, and conducted a comprehensive evaluation of 54 major and important disease (including several most deadly cancers) diagnosis tasks across 15 main anatomies. Experimental results demonstrate the substantial potential of fVLM in versatile medical image interpretation. In the zero-shot classification task, we achieved an average AUC of 81.3% on 54 diagnosis tasks, surpassing CLIP and supervised methods by 12.9% and 8.0%, respectively. Additionally, on the publicly available CT-RATE and RadChestCT benchmarks, our fVLM outperformed the current state-of-the-art methods with absolute AUC gains of 7.4% and 4.8%, respectively. Code is available at https://github.com/alibaba-damo-academy/fvlm

## 1 INTRODUCTION

Medical image interpretation is a critically important yet exceptionally burdensome task in clinical workflows, particularly when dealing with 3D imaging scans Udare et al. (2022). Radiologists are required to examine hundreds of slices across dozens of anatomies meticulously Blankemeier et al. (2024). As a result, there is a growing demand for versatile and reliable AI to assist in the automated interpretation of medical images for a wide range of diagnostic needs. Supervised learning is a prominent strategy for automating this process, demonstrating remarkable success in natural scene

---

*Correspondence to Jianpeng Zhang. The work was done during Zhongyi's internship at DAMO Academy

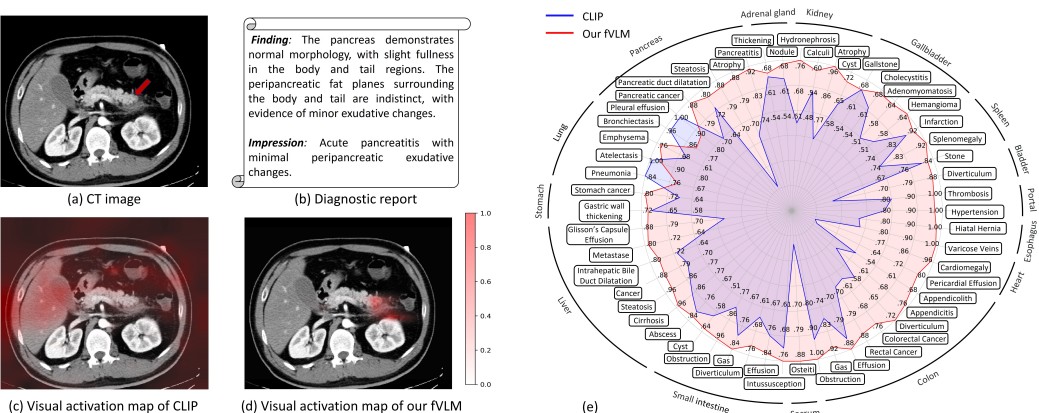

Figure 1: Comparative analysis of vanilla VLM (CLIP) and our fine-grained VLM (fVLM). (a,b) A representative CT slice and its corresponding radiological report. (c,d) Visual activation maps generated by CLIP and fVLM respectively, illustrating regions of interest for pancreatitis diagnosis. (e) Quantitative comparison of AUC scores across 54 disease diagnosis tasks in 15 anatomies.

images, such as ImageNet Deng et al. (2009). In the medical domain, specific disease category information must be precisely defined in advance, necessitating extensive annotations from specialized annotators Isensee et al. (2021); Wang et al. (2023); Zhang et al. (2023a); Guo et al. (2024). Unlike natural images, medical images encompass a complex variety of conditions, making it challenging to fulfill all clinical diagnostic requirements through a predefined one-hot label space Liu et al. (2023c). Furthermore, the labor-intensive annotation constitutes an additional burden on doctors outside of their regular duties. These challenges make it particularly difficult to apply supervised learning methodologies effectively within the medical field.

Recently, Vision Language Models (VLMs) Zhang et al. (2022); Tiu et al. (2022); Lin et al. (2023); Wu et al. (2023); Blankemeier et al. (2024) have gained considerable attention, presenting a promising alternative to supervised learning paradigms. The fundamental concept involves supervising model training directly through diagnostic reports, thus eliminating the need for specific disease category labels Cao et al. (2024). Radiology reports are highly condensed recordings of the diagnostic process, meticulously documenting the evaluations conducted by at least one experienced radiologist. During this evaluation, they can reference patient history and clinical information, resulting in a text-based annotation. Current VLMs predominantly employ global contrastive learning, wherein embeddings of entire images and reports from the same patient are brought closer together, while those from different patients are pushed apart Bai et al. (2024); Hamamci et al. (2024). However, this global contrast is inherently coarse-grained, overlooking local similarities or disparities between anatomical regions and report sentences. Pulling certain anatomical regions closer to unrelated text or vice versa may result in misleading alignment, making it challenging to align complex medical images and reports within a unified representation space. As illustrated in Fig. 1 (c), the attention map of the CLIP Radford et al. (2021), a vanilla coarse-grained VLM, is visualized when executing certain diagnostic tasks. The result demonstrates that such a global alignment mechanism can readily induce the model to focus on the regions that are not relevant to the diagnosis, potentially compromising its performance and interpretability.

In this paper, we propose a fine-grained vision-language model (fVLM) for automated CT image interpretation. This model moves beyond the traditional global image-text contrastive learning pipeline, enabling anatomy-level fine-grained alignment between CT scans and reports. Our motivation arises from the fact that diagnostic reports typically document clinically significant abnormal findings in various organs or body structures in the CT images per anatomy level, thus establishing an intrinsic fine-grained vision-language correspondence between any text-described finding and its image location. Specifically, we perform anatomical-level decomposition and matching for both the images and reports, followed by fine-grained alignment of the matched visual embeddings and the corresponding report embeddings of the same anatomy. This explicit matching alleviates the misalignment issues associated with global contrastive learning and enhances the interpretability of VLMs, as illustrated in Fig. 1 (d). Moreover, fine-grained alignment encounters significant chal-

lenges related to false negatives, primarily arising from the prevalence of anatomy-level healthy samples and similar abnormalities across different diseases, which could result in ambiguous pairings at the patient level. We introduce a simple yet effective method to identify and manage the massive false negatives from both normal and abnormal samples, and advocate for a shift in contrastive learning from a broad patient-level pairing to a more nuanced disease-aware pairing approach.

Due to privacy concerns and the scarcity of quality medical data, the limited availability of vision-language data has been one of the most significant bottlenecks for medical VLMs. To overcome this limitation, we have curated the largest CT dataset to date, named MedVL-CT69K, which includes 272,124 CT scans from 69,086 unique patients and their corresponding diagnostic reports. On this extensive dataset, our fVLM has demonstrated outstanding zero-shot diagnostic capabilities, achieving an average AUC of 81.3% across 54 disease diagnosis tasks, surpassing the competing CLIP model by 12.9% (see Fig. 1 (e)) and the supervised baseline by 8.0%. Moreover, on the publicly available CT-RATE and Rad-ChestCT datasets, our fVLM outperforms the state-of-the-art approach by 7.4% and 4.8% absolute AUC value gains, respectively. Beyond diagnostic tasks, the model also exhibits remarkable proficiency in downstream report-generation tasks. Our key contributions are summarized as follows:

1. We propose a scalable and annotation-free vision-language model, fVLM, for CT image interpretation, which demonstrates strong scaling capabilities to meet a wide range of clinical diagnostic needs.

2. We address the vision-language misalignment issues of VLMs by employing a fine-grained anatomy-level contrastive learning framework.

3. We introduce a dual false negative reduction module to alleviate the adverse effects of false negatives in both normal and abnormal samples.

4. Extensive experiments on a large-scale in-house dataset as well as two public benchmarks demonstrate the advantages of fVLM over the state-of-the-art counterparts.

## 2 RELATED WORK

### 2.1 MEDICAL VISION-LANGUAGE PRE-TRAINING

Existing medical vision-language pre-training (Med-VLP) methods primarily focus on 2D images depicting a single body part, notably chest X-rays (CXR). Most of them learn transferable representations by aligning the medical scans and corresponding reports with contrastive loss Zhang et al. (2022); Boecking et al. (2022); Tiu et al. (2022); Huang et al. (2023); Zhou et al. (2023); Zhang et al. (2023b); Lin et al. (2023); Liu et al. (2023b); Bannur et al. (2023); Cheng et al. (2023); Liu et al. (2023a); Lin et al. (2023); Sun et al. (2024); Lu et al. (2024); Christensen et al. (2024). In particular, MedKLIP Wu et al. (2023) and KAD Zhang et al. (2023c) utilize medical domain knowledge to enhance the textual information extraction, thereby improving the contextual understanding of radiology reports. Imitate Liu et al. (2023b) derives multi-level visual features from CXR images and separately aligns these features with descriptive and conclusive text in hierarchical medical reports. Given the paucity of paired image-text data in the medical domain, several studies have investigated data-efficient Med-VLP. Notably, MedCLIP Wang et al. (2022b) and PTUnifier Chen et al. (2023) use unpaired CXR images and reports for multimodal pre-training. Pairaug Xie et al. (2024) designs a pairwise augmentation approach that scales up the training data by manipulating existing image-report pairs or generating entirely new cases. Beyond inspecting a single body part, recent studies have expanded the scope of VLP to encompass broader anatomical structures within high-detail 3D CT images Cao et al. (2024); Hamamci et al. (2024); Bai et al. (2024); Lin et al. (2024); Blankemeier et al. (2024), enabling more comprehensive diagnostic support in clinical practice. Specifically, BIUD Cao et al. (2024) and CT-CLIP Hamamci et al. (2024) align chest CT volumes and radiology reports. Merlin Blankemeier et al. (2024) focuses on abdomen scenarios and incorporates structured electronic health record (EHR) data as additional supervision.

While existing Med-VLP studies have demonstrated decent performance, they predominantly employ a global alignment scheme that contrasts entire images and reports Zhang et al. (2022); Tiu et al. (2022), overlooking the local similarities or disparities between image patches and report pieces. This oversight can result in a misalignment problem Müller et al. (2022), constraining the

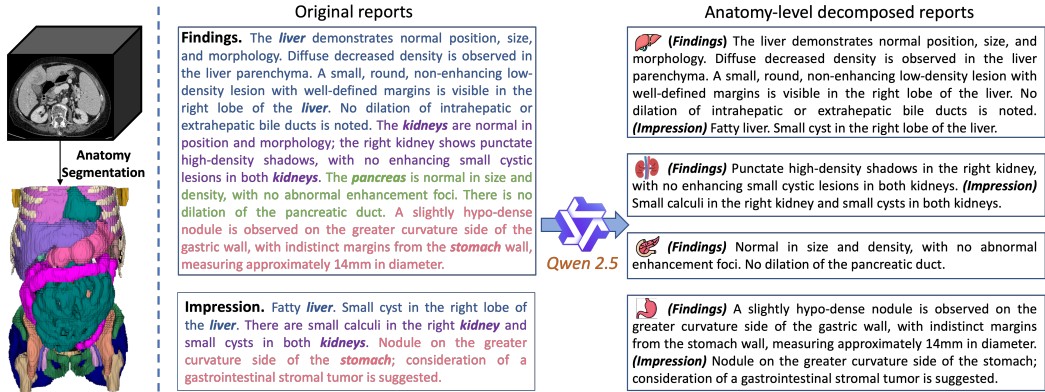

Figure 2: Illustration of CT anatomy parsing (left) and diagnostic report decomposition (right).

model to a coarse-grained understanding and limiting its capacity to capture fine-grained, clinically relevant details.

## 2.2 FINE-GRAINED ALIGNMENT IN MED-VLP

To address the misalignment challenge, GLoRIA Huang et al. (2021), LoVT Müller et al. (2022) and MGCA Wang et al. (2022a) integrate global contrastive learning with a local alignment technique. They leverage a cross-attention mechanism to implicitly learn fine-grained correspondences between image regions and report sentences within each sample. However, while this implicit local alignment has demonstrated effectiveness for 2D CXR data, we argue that its applicability to 3D CT volumes may be limited due to the dramatically higher data complexity. Specifically, compared to 2D CXR images that involve only a few anatomical anatomies Li et al. (2024b), 3D CT scans typically encompass hundreds of anatomical structures and provide detailed, volumetric views of the human body Wasserthal et al. (2023). This increased imaging complexity enables a deeper analysis of intricate medical conditions while concurrently yielding more extensive and nuanced radiology reports that delineate wide-ranging anatomical features and clinical findings Udare et al. (2022); Blankemeier et al. (2024). Given these distinctions, the endeavor to learn local alignments implicitly, which is already prone to be sensitive to hyper-parameters and difficult to train Müller et al. (2022), becomes exceedingly intractable in CT scenarios.

## 3 METHOD

### 3.1 DATA PRE-PROCESSING

**Anatomy parsing.** We utilize Totalsegmentator to generate detailed anatomical structure masks for 104 regions within CT scans Wasserthal et al. (2023), encompassing organs, bones, muscles, and vessels, as illustrated in Fig. 2. Subsequently, we group these 104 regions into 36 major anatomies to align with the granularity of descriptions in clinical reports, as detailed in Appendix Tab. 6. This grouping is necessary because CT diagnosis reports often lack precise localization of the lesion areas Li et al. (2024a). For instance, the lung is segmented into five distinct lobes in Totalsegmentator Wasserthal et al. (2023), while a report might merely state "lung inflammation" without specifying which lobe is affected. This ambiguity presents a significant challenge in precisely extracting corresponding diagnostic descriptions for each lobe from the report. Furthermore, even when the lesion locations are reported in some cases, the probability of anomalies occurring at a specific fine-grained anatomical site (*i.e.*, right middle lobe) is considerably low, leading to an overwhelming imbalance between normal and abnormal samples for that anatomical structure. As a result, most mini-batches may consist entirely of normal samples, which may skew the training process and impair the model's diagnostic capability. Overall, anatomical grouping entails a trade-off among analytical granularity, image-text consistency, and data balance.

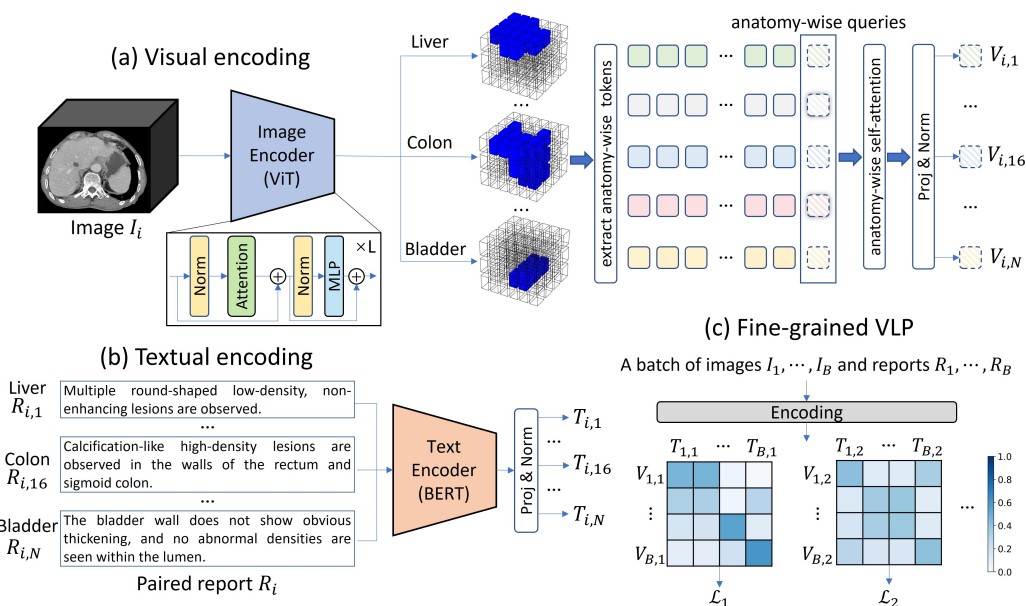

Figure 3: Framework of fVLM. (a) Visual encoding. We input a CT volume $I_i$ into the image encoder and extract corresponding visual tokens for each anatomy. We then append an anatomy-specific query token to the extracted visual tokens of each anatomy. These query tokens are subsequently updated through self-attention, constituting the visual representations of their respective anatomies. $N$ is the number of anatomies. (b) Textual encoding. We decompose the paired report $R_i$ into anatomy-wise descriptions and feed them separately into the text decoder to obtain anatomy-specific textual representation. (c) Fine-grained VLP. We perform local alignment for each individual anatomy across different CT scans. $L_j$ denotes the contrastive loss computed for the *j*-th anatomy.

**Report decomposition.** As depicted in Fig. 2, we decompose raw CT diagnostic reports according to the grouped anatomies. To reduce the complexity, we employ a divide-and-conquer strategy, executing the decomposition process for the *findings* and *impression* sections of each report independently, followed by an integration of extracted anatomy-level descriptions. Our approach is delineated in the following three steps. First, we design a prompt (see Appendix Fig. 6) and employ the LLM, Qwen 2.5 Bai et al. (2023), to identify all anatomies mentioned in both sections. Notably, we found that when one section lacks explicit references to some anatomies but instead mentions their anatomical sub-structures or uses medical terminology as referents, the LLM may fail to recognize these anatomies due to insufficient domain knowledge. To mitigate these potential omissions, we employ a complementary string-matching strategy. For instance, the inclusion of terms such as *"jejunum"*, *"ileum"*, or *"duodenum"* in the section will prompt the recognition of *"small intestine"*. Second, we use the LLM to extract anatomy-specific descriptions from both sections, with the prompt detailed in Appendix Fig. 7. Lastly, a simple post-processing is performed to integrate the anatomy-level descriptions extracted from these two sections. Specifically, for each anatomy mentioned in both sections, we concatenate the extracted *findings* content with its corresponding *impression* description. In instances where the anatomy appears in only one section, we supplement the absent component with a *"null"* string before concatenation. If one anatomy is not mentioned in either section, we default its description to *"{anatomy} shows no significant abnormalities."* based on established clinical practice.

## 3.2 FINE-GRAINED CONTRASTIVE PRE-TRAINING

Our approach is grounded in the CLIP architecture Radford et al. (2021), which aligns visual and linguistic modalities through contrastive learning of positive and negative pairs. Following Bai et al. (2024); Cao et al. (2024); Lu et al. (2024), we adopt vision transformer (ViT) Dosovitskiy et al. (2020) and BERT Devlin et al. (2018) as the image and text encoder, respectively. Given a CT

volume $I_i \in \mathbb{R}^{1 \times D \times H \times W}$, where $D$, $H$ and $W$ represent the inter-slice, spatial height and width dimensions respectively, the vision encoder transforms the input into a compact visual embedding $\mathcal{F}_i \in \mathbb{R}^{c \times d \times h \times w}$. For each anatomy, we utilize its segmentation mask $M_{i,j} \in \{0,1\}^{D \times H \times W}$, where 0 represents the background and 1 denotes the foreground, to guide the construction of anatomy-specific visual representations. Specifically, we begin by partitioning $M_{i,j}$ into non-overlapping patches of size $\frac{D}{d} \times \frac{H}{h} \times \frac{W}{w}$. Each patch spatially corresponds to a visual token in $\mathcal{F}_i$. Then, we locate the patches that contain foreground elements of $M_{i,j}$ and extract their associated tokens as the visual descriptors of the *j*-th anatomy. Next, we append a learnable anatomy-wise query token to these extracted tokens and update it via a self-attention layer. Finally, the updated query token is fed into a linear projection layer followed by L2-normalization to generate anatomy-wise visual representation $V_{i,j}$. Given the irregular sizes of CT images between patients, we employ RandomCrop to facilitate the construction of mini-batches. It is important to note that anatomies truncated by the cropping operation will be overlooked to maintain the integrity of anatomical visual content during contrastive alignment. The absence of this visual information could include critical diagnostic cues, potentially resulting in alignment failures.

Given the image's associated report $R_i$, we decompose it into discrete descriptions $R_{i,j}$ for each anatomy, as detailed in Sec. 3.1. Then, we employ the text encoder to transform $R_{i,j}$ into anatomy-specific textual embeddings $T_{i,j}$. For a mini-batch of images $\{I_1, \cdots, I_B\}$ and reports $\{R_1, \cdots, R_B\}$, we calculate the softmax-normalized image-to-text and text-to-image similarity as:

$$p_{i,j,k}^{i2t} = \frac{e^{\langle V_{i,j}, T_{k,j}\rangle / \tau}}{\sum_{k'=1}^{N_j} e^{\langle V_{i,j}, T_{k',j}\rangle / \tau}}, \quad p_{i,j,k}^{t2i} = \frac{e^{\langle T_{i,j}, V_{k,j}\rangle / \tau}}{\sum_{k'=1}^{N_j} e^{\langle T_{i,j}, V_{k',j}\rangle / \tau}} \tag{1}$$

where $j$ denotes anatomy index, $N_j$ is the number of structurally complete samples for the *j*-th anatomy after RandomCrop, $\langle a, b\rangle$ refers to the cosine similarity between vectors $a$ and $b$, $\tau$ is a learnable temperature parameter. The total loss is computed as:

$$L_{itc} = \frac{1}{2} \left( \sum_{j=1}^{T} \frac{1}{N_j} \sum_{i=1}^{N_j} \left( \mathrm{H}(y_{i,j}^{i2t}, p_{i,j}^{i2t}) + \mathrm{H}(y_{i,j}^{t2i}, p_{i,j}^{t2i}) \right) \right) \tag{2}$$

in which $T$ is the number of anatomy categories, and H is cross-entropy loss. $y_{i,j}^{i2t}$ and $y_{i,j}^{t2i}$ denote ground-truth one-hot similarity, where negative pairs have a probability of 0 and the positive pair has a probability of 1.

## 3.3 REDUCING FALSE NEGATIVES IN IMAGE-REPORT PAIRS

The core of contrastive-based VLP lies in instance-level pairing, which brings together the vision and language modalities of the same instance while distancing different instances. However, there are often complex semantic relationships between different instances (patients) in medical contexts Hamamci et al. (2024). For example, patients diagnosed as normal are semantically consistent and abnormal samples with the same pathologies also exhibit high semantic similarities. These semantically similar samples constitute false negatives when they co-occur within the same mini-batch during contrastive pre-training, and inadvertently increasing their distances could degrade the diagnostic accuracy of medical VLMs. To address this issue, we propose a dual false negative reduction (FNR) approach that goes beyond instance-level pairing and pursues a more comprehensive understanding of the semantic landscape in medical imaging.

When performing global contrastive learning between entire images and reports Hamamci et al. (2024), a patient sample is diagnosed as normal only if all scanned anatomies are free of abnormalities. Under this definition, the proportion of normal samples is notably low (*e.g.*, 0.2% in MedVL-CT69K). However, in our fine-grained framework, the number of normal cases increases substantially (refer to Appendix Fig. 8) due to the more granular definition of normality at the anatomy level: although a CT examination reveals abnormalities in specific anatomical structures, those unaffected anatomies can still be considered normal based on established clinical protocols. This substantial increase in normal samples leads to a proliferation of false negatives in our fine-grained contrastive learning framework. Moreover, in contrast to the relatively fixed template-style descriptions for entirely normal images Cao et al. (2024), we observe considerable variability in the descriptions for normal cases of each anatomy. As a result, how to identify and cope with these massive normal samples poses a critical challenge in unlocking the full potential of our method. We address this issue by

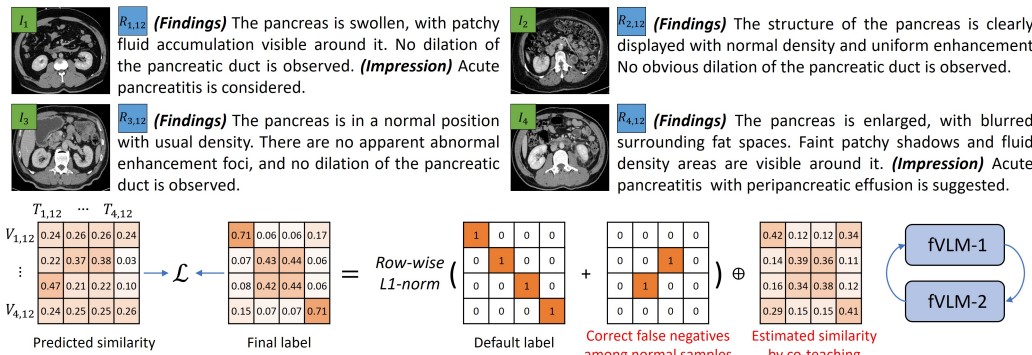

Figure 4: Illustration of the proposed dual false negative reduction approach. $V_{i,12}$ and $T_{i,12}$ represent pancreas-specific visual and textual features from the $i$-th sample, respectively. On one hand, we identify those samples not mentioned in the *impression* section of reports as normal and correct the corresponding labels for these semantically consistent samples to 1 in the label matrix. On the other hand, we further incorporate the estimated image-text similarities into the label matrix, aiming to capture potential semantic relationships between different samples and thereby enhance the model's semantic comprehension. Notably, to mitigate error accumulation in this process, we propose a co-teaching strategy, wherein two fVLMs are trained alternately and the image-text similarities employed by one model are estimated from the other.

leveraging the inherently hierarchical structure of CT reports Hamamci et al. (2024); Blankemeier et al. (2024). Specifically, the *findings* section of a report outlines all observations derived from the image, including the appearance of anatomical structures, any abnormalities such as masses or lesions, and the conditions of various anatomical components. Meanwhile, the *impression* section consolidates the abnormal observations into a concise summary, offering standard diagnostic conclusions and highlighting potential diseases. Based on this prior, we empirically annotate anatomies not mentioned in the *impression* section as normal. We then correct $y_{i,j,k}^{i2t}$ and $y_{i,j,k}^{t2i}$ to 1 if the $i$-th and $k$-th patients are both normal in terms of the $j$-th anatomy. Moreover, to stabilize model training, we normalize $y_{i,j}^{i2t}$ and $y_{i,j}^{i2t}$ so that their sums equal 1.

Furthermore, due to the narrowed abnormality space from the entire body to specific anatomical regions, the abnormal samples in our fine-grained framework typically exhibit higher semantic similarity compared to those in global contrastive learning methods. For instance, although two patients exhibit significant overall differences, their pancreas may manifest the same pathology as described as follows: $R_{1,12}$: *"The pancreas is swollen, with patchy fluid accumulation visible around it. Acute pancreatitis is considered"* and $R_{2,12}$: *"The pancreas is enlarged, and fluid density shadows are visible around it. Acute pancreatitis with peripancreatic fluid collection is suggested."* Here, the subscript 12 denotes the index of the pancreas in all involved anatomies. In this scenario, pushing away the visual features $V_{1,12}$ from the textual features $T_{2,12}$ is unreasonable and potentially compromises the model's capability in pancreatitis diagnosis. To accommodate this heightened inter-sample similarity, we propose a bootstrapping strategy that utilizes the similarity scores $p^{i2t}$ and $p^{t2i}$ predicted by the model itself to dynamically correct the target label during contrastive pre-training. However, these predicted similarity scores may be biased. Further incorporating them into model training could cause error accumulation and ultimately result in significant performance degradation. To tackle this, we propose a co-teaching training framework that alternately trains two fVLMs, where the image-text similarity scores predicted by one model are used to correct the contrastive learning target of the other model:

$$
\begin{aligned}
\boldsymbol{y}^{i2t} = \alpha \boldsymbol{y}^{i2t} + (1-\alpha)\boldsymbol{p}^{i2t\prime}, \quad \boldsymbol{y}^{i2t\prime} = \alpha \boldsymbol{y}^{i2t\prime} + (1-\alpha)\boldsymbol{p}^{i2t} \\
\boldsymbol{y}^{t2i} = \alpha \boldsymbol{y}^{t2i} + (1-\alpha)\boldsymbol{p}^{t2i\prime}, \quad \boldsymbol{y}^{t2i\prime} = \alpha \boldsymbol{y}^{t2i\prime} + (1-\alpha)\boldsymbol{p}^{t2i}
\end{aligned}
\tag{3}
$$

Here $\alpha \in [0,1]$ is a free parameter and we empirically set it to 0.5 in this work. $\boldsymbol{p}^{i2t\prime}$ and $\boldsymbol{p}^{t2i\prime}$ are predicted image-to-text and text-to-image similarities from another model. To reduce the risk of concurrent errors arising from both models for the same image-text pair, we enhance their diversity

Table 1: Zero-shot performance comparison on the MedVL-CT69K dataset. The best and second-best zero-shot results are highlighted in **bold** and underlined.

| | Method | AUC | ACC | Spec | Sens | F1 | Prec |
|---|---|---|---|---|---|---|---|
| | Baseline | 73.3 | 69.1 | 76.2 | 62.0 | 79.4 | 17.6 |
| Supervised | CT-VocabFine | 76.7 | 72.2 | 76.1 | 68.2 | 81.6 | 20.3 |
| | CT-LiPro | 76.5 | 70.9 | 76.8 | 65.1 | 81.3 | 19.3 |
| | CLIP Radford et al. (2021) | 68.4 | 66.7 | 68.0 | 65.5 | 76.0 | 18.0 |
| | LOVT Müller et al. (2022) | 69.4 | 65.4 | 60.1 | 70.8 | 70.9 | 15.2 |
| | MGCA Wang et al. (2022a) | 70.1 | 66.4 | 64.5 | 68.3 | 73.9 | 16.0 |
| | Imitate Liu et al. (2023b) | 70.6 | 67.9 | 66.6 | 69.2 | 75.4 | 17.9 |
| Zero-shot | ASG Li et al. (2024a) | 70.1 | 67.7 | 67.4 | 68.0 | 75.9 | 18.8 |
| | CT-GLIP Lin et al. (2024) | 69.3 | 66.9 | 63.1 | 70.6 | 74.2 | 18.1 |
| | BIUD Cao et al. (2024) | 71.4 | 69.2 | 69.0 | 69.3 | 76.6 | 18.7 |
| | Merlin Blankemeier et al. (2024) | 71.9 | 69.5 | 69.7 | 69.2 | 77.0 | 18.1 |
| | Ours | **81.3** | **76.2** | **76.5** | **75.8** | **82.2** | **21.1** |

by employing different model initialization, data iteration sequences and augmentations. Fig. 4 exemplifies the calculation of the final labels.

## 4 EXPERIMENTS

### 4.1 EXPERIMENTAL SETUP

**Dataset.** In this study, we curate MedVL-CT69K, a large-scale CT dataset comprising 272,124 CT scans from 69,086 unique patients and their associated reports. Each patient consists of a non-contrast CT scan and contrast-enhanced CT scans, which include one or more of the following phases: arterial, venous, and delayed. We randomly split the dataset into training, validation and test sets of 64,476, 1,151, and 3,459 patients, respectively. The validation and test sets are annotated with 36 and 54 diseases by expert radiologists. A detailed distribution of these diseases is provided in Appendix Tab. 9 and Tab. 10. Additionally, we conduct experiments on two benchmarks, CT-RATE Hamamci et al. (2024) and Rad-ChestCT Draelos et al. (2021). Following Hamamci et al. (2024), we train fVLM on the training set of CT-RATE and use its test set and the whole Rad-ChestCT dataset for internal and external evaluations, respectively. The details regarding these two datasets can be found in Hamamci et al. (2024) and Draelos et al. (2021).

**Evaluation metrics.** We compare the performance of different pre-training methods on zero-shot abnormality detection and downstream report-generation tasks. For the zero-shot experiments, following Hamamci et al. (2024), we adopt the area under the ROC curve (AUC), balanced accuracy (ACC), specificity (Spec), sensibility (Spec), precision (Prec) and weighted F1-score as the metrics. For the report-generation task, we employ both diagnostic metrics and natural language generation metrics for model evaluation. To facilitate the calculation of diagnostic metrics, we develop a high-performing text classifier to identify abnormalities in generated radiology reports. A detailed exposition of the classifier's training and evaluation is provided in Appendix A.1.

**Implementation details** are available in Appendix A.2.

### 4.2 ZERO-SHOT ABNORMALITY DETECTION

Through the extensive MedVL-CT69K dataset, we compare the zero-shot abnormality detection performance of different methods on 54 diseases across 15 anatomies. The results are presented in Tab. 1. It can be seen that our method outperforms all counterparts by a large margin. Specifically, it suppresses CLIP Radford et al. (2021) by 12.9 points on AUC and 9.5 points on ACC. Furthermore, compared to the second-best competitor, Merlin Blankemeier et al. (2024), our method achieves absolute gains of 9.4 points on AUC and 6.7 points on ACC. Notably, we observe that LOVT Müller et al. (2022) and MGCA Wang et al. (2022a) exhibit marginal performance improvements over CLIP, which underscores the significant limitations of implicit local alignment methodologies in CT

Table 2: Performance comparison on the CT-RATE and Rad-ChestCT benchmarks. Here, CT-CLIP refers to the CLIP model trained on the CT-RATE dataset, as named in the original paper. The best and second-best zero-shot results are highlighted in **bold** and underlined.

| Dataset | Metric | Supervised | | | Zero-shot | | | |
|---|---|---|---|---|---|---|---|---|
| | | Baseline | CT-VocabFine | CT-LiPro | CT-CLIP | BIUD | Merlin | Ours |
| Internal validation (CT-RATE) | AUC | 60.3 | 75.0 | 75.1 | 70.4 | 71.3 | 72.8 | **77.8** |
| | ACC | 58.1 | 69.2 | 67.6 | 65.1 | 68.1 | 67.2 | **71.8** |
| | F1 | 63.2 | 72.8 | 71.4 | 69.1 | 71.6 | 70.9 | **75.1** |
| | Prec | 24.0 | 34.2 | 33.1 | 30.6 | 33.8 | 33.7 | **37.9** |
| | Spec | - | - | - | - | 68.6 | 66.8 | **71.7** |
| | Sens | - | - | - | - | 67.3 | 70.1 | **72.8** |
| External validation (Rad-ChestCT) | AUC | 54.1 | 64.9 | 64.7 | 63.2 | 62.9 | 64.4 | **68.0** |
| | ACC | 53.9 | 62.2 | 62.5 | 59.9 | 60.6 | 61.9 | **64.7** |
| | F1 | 58.7 | 66.5 | 66.8 | 64.8 | 65.2 | 66.3 | **68.8** |
| | Prec | 28.7 | 35.9 | 35.3 | 33.9 | 33.7 | 34.8 | **37.4** |
| | Spec | - | - | - | - | 60.2 | 61.7 | **64.6** |
| | Sens | - | - | - | - | 59.6 | 61.0 | **64.6** |

Table 3: Performance comparison on the downstream report-generation task using the MedVL-CT69K dataset. 'IN' means initialization with ImageNet supervised weights, while all other methods are trained with our dataset. 'SP' denotes the supervised baseline model.

| Encoder | Init | ACC | GREEN | BLEU-1 | BLEU-2 | BLEU-3 | BLEU-4 | METEOR | ROUGE-L | CIDEr |
|---|---|---|---|---|---|---|---|---|---|---|
| Frozen | IN | 51.3 | 34.0 | 47.4 | **32.2** | **25.5** | **21.2** | 28.1 | 44.3 | 10.6 |
| | SP | 55.8 | 25.9 | 48.3 | 26.3 | 18.0 | 12.8 | 30.8 | 40.6 | 6.6 |
| | MAE | 50.7 | 21.6 | 49.0 | 27.1 | 18.6 | 13.1 | 30.5 | 41.6 | 6.1 |
| | CLIP | 57.3 | 33.4 | 49.7 | 28.8 | 20.7 | 15.5 | 31.0 | 42.2 | 9.6 |
| | BIUD | 58.4 | 33.7 | 47.1 | 30.4 | 23.4 | 18.9 | 29.1 | 44.2 | 13.9 |
| | Merlin | 58.8 | 34.2 | 49.5 | 31.4 | 23.9 | 19.0 | 30.0 | 43.8 | 14.3 |
| | Ours | **61.5** | **37.2** | **50.7** | **32.2** | 24.5 | 19.6 | **31.3** | **45.1** | **14.9** |
| Fine-tuning | IN | 58.0 | 33.4 | 49.4 | 29.8 | 21.9 | 16.9 | 30.4 | 43.6 | 10.0 |
| | SP | 60.6 | 35.5 | 49.9 | 30.7 | 22.9 | 17.9 | 30.6 | 43.4 | 11.7 |
| | MAE | 54.4 | 29.4 | 49.0 | 27.1 | 18.6 | 15.1 | 30.5 | 42.5 | 8.8 |
| | CLIP | 62.0 | 37.6 | 49.9 | 31.9 | 24.3 | 19.5 | 30.7 | 44.7 | 14.3 |
| | BIUD | 62.6 | 38.8 | 50.2 | 31.7 | 24.0 | 19.0 | 30.9 | 44.7 | 13.8 |
| | Merlin | 63.0 | 39.2 | 50.7 | 33.3 | 25.8 | 20.9 | 31.1 | 46.0 | **17.2** |
| | Ours | **64.5** | **40.2** | **52.2** | **34.5** | **26.8** | **21.9** | **31.6** | **46.4** | 17.1 |

imaging scenarios. We enumerate the detection performance of our model for each abnormality in Appendix Tab. 11.

Tab. 2 exhibits the performance comparison with the current state-of-the-art method (*i.e.*, CT-CLIP) on the CT-RATE and Rad-ChestCT benchmarks. In the zero-shot setting, our method demonstrates significant improvements over CT-CLIP, achieving absolute AUC gains of 7.4% and 4.8% in the internal and external evaluations, respectively. Notably, the zero-shot performance of our model even outperforms the outcomes of CT-VocabFine and CT-LiPro that are both derived from CT-CLIP through supervised fine-tuning. To be specific, in the internal test set, our approach exceeds CT-VocabFine and CT-LiPro by 2.3 and 3.7 points on F1-score. In the external test set, it surpasses these two models by 2.3 and 2.0 points on F1-score.

To further assess the clinical utility of our method, we conduct a reader study to compare our method with three board-certified radiologists. Please refer to Appendix A.3 for detailed results and discussions.

## 4.3 RADIOLOGY REPORT GENERATION

To assess the transfer abilities of VLMs, we conduct experiments on the downstream task of radiology report generation using the MedVL-CT69K dataset. For these experiments, we integrate each pre-trained image encoder with a BERT-base text decoder for whole report generation. The generation process is optimized using the language modeling loss Devlin et al. (2018).

Table 4: Effect of our proposed modules. CLIP serves as the baseline.

| FGA | FNCN | CoT | AUC | ACC |
|-----|------|-----|-----|-----|
|     |      |     | 70.9 | 69.3 |
| ✓   |      |     | 76.0 | 74.0 |
| ✓   | ✓    |     | 78.7 | 75.3 |
| ✓   |      | ✓   | 77.5 | 74.6 |
| ✓   | ✓    | ✓   | 79.8 | 75.9 |

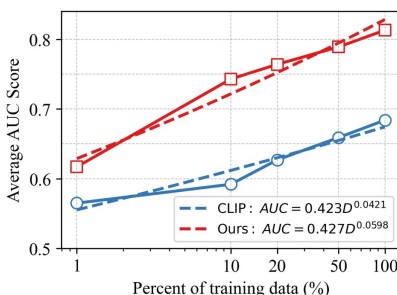

Figure 5: Scaling laws of CLIP and our method.

Tab. 3 presents the experimental results in both frozen and fine-tuning protocols, where the frozen protocol keeps the pre-trained image encoder fixed, while the fine-tuning protocol allows the entire model to be updated during training. It demonstrates that vision-language pre-trained models outperform those with purely visual pre-training, underscoring the benefits of aligning visual and textual features into a unified representation space for the report generation task. Notably, in the frozen regime, our method significantly outperforms CLIP by 4.2 points on ACC and 3.8 points on GREEN Ostmeier et al. (2024). While the performance gap attenuates in the fine-tuning protocol, our method still surpasses CLIP by a clear margin, achieving a 2.5-point improvement on ACC and and a 2.6-point improvement on GREEN. Although the results have demonstrated the superiority of our approach, we argue that directly employing the fine-grained alignment model for whole report generation may not unleash its full power due to the granularity mismatch issue. We will explore a potentially more effective strategy of generating anatomy-wise diagnostic reports in our future work.

### 4.4 ANALYSIS OF OUR FRAMEWORK

**Ablation study.** We investigate the impact of three modules on the performance of fVLM on the validation set of MedVL-CT69K, including **f**ine-**g**rained **a**lignment (FGA), **f**alse **n**egatives **c**orrection between **n**ormals (FNCN) and **co-t**eaching strategy (CoT). As shown in Tab. 4, each enhancement component contributes to the improvement of the model's performance. Notably, the FGA and FNCN contribute the largest performance gains. The combination of them leads to an overall improvement of 7.8 points on AUC and 6.0 points on ACC. Furthermore, in Appendix A.4, we demonstrate that applying CoT to correct contrastive learning labels yields superior results compared to using either the training model or the momentum model.

**Scaling law.** In Fig. 5, we compute the data scaling law curves to assess how the performance of CLIP and our method improves as the volume of training data increases. It can be seen that our approach consistently outperforms CLIP across multiple data scales, exhibiting superior data efficiency.

**Visualization analysis.** The visualization results and discussions can be found in Appendix A.5

## 5 CONCLUSION

In this paper, we have presented fVLM, a fine-grained vision-language pre-training method for CT data. Our proposed methodology explicitly aligns discrete anatomical structures in CT scans with their corresponding descriptions in diagnostic reports, thereby addressing the misalignment issue of CLIP and its existing variants that contrast entire images and reports. Extensive experiments, including quantitative abnormality detection and report generation tasks as well as qualitative visualization analysis, demonstrate the superiority of fVLM.

**Limitations and future work.** The implementation of our fine-grained alignment methodology necessitates localizing anatomical structures in CT images and decomposing diagnostic reports into anatomy-wise sub-descriptions. This data processing step entails additional resource consumption and time commitment. For future work, we plan to investigate anatomy-wise report generation to fully unleash the potential of fVLM on this application.

**Acknowledgements.** Jianpeng Zhang was supported by the Zhejiang Province Postdoctoral Research Excellence Funding Program (ZJ2024032). This work was also supported by the Zhejiang Province "Pioneer Eagle + X" Research and Development Initiative (2024C03043).

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
