# A    APPENDIX

{*findings or impression*}

You are a professional radiologist. Please determine if the anatomy ({*anatomy*}) is mentioned in this CT image report. Please answer directly with "Yes" or "No".

Figure 6: Prompt used to judge if an anatomy is mentioned in the "Findings" or "Impression" section of a clinical report.

{*findings or impression*}

You are a professional radiologist. Please extract the descriptive information about the specific anatomy ({*anatomy*}) from this CT image diagnostic report. Please follow these guidelines:

**Precise extraction:** Extract the descriptive information directly related to {*anatomy*} from the report.
**Specify anatomical details:** If the report mentions specific areas, parts, or anatomical details of {*anatomy*}, make sure to include this information in the description. This could include affected areas, normal structures, or any notable features.
**Concise and clear:** Directly extract the report content, avoiding unnecessary explanations or background.
**Format requirement:** Please provide the information in the format "{*anatomy*}: descriptive information", ensuring to use {*anatomy*} as the unified prefix for the item. Even if the anatomy has multiple independent parts or multiple lateral characteristics, it should be treated and described as a whole, returning only one comprehensive piece of information about that anatomy.

Figure 7: Prompt used to extract anatomy-specific description from the "Findings" or "Impression" section of a clinical report. Notably, we do not obtain the descriptions for all anatomies in a single query; rather, we strategically query the LLM for each anatomy individually. This approach significantly simplifies the complexity of description extraction and greatly enhances the quality of the extracted descriptions.

## A.1    DETAILS ABOUT THE TEXT CLASSIFIER

We utilize the annotated validation and test sets of MedVL-CT69K to develop a text classifier that identifies 54 abnormalities in the generated radiology reports. To achieve this, we first merge these two sets and then re-split them into new training and validation sets using a 2:1 ratio. Afterwards, we train the classifier, which consists of a BERT-base encoder and a classification head, using the reports and corresponding disease labels form the training set. A binary cross-entropy loss is used to supervise the model training. Tab. 7 shows the precision, recall, and F1 scores of the text classifier across 54 abnormalities on the validation set. Notably, the model achieves an impressive average F1 score of 0.95. This high performance substantiates its reliability as a tool for assessing the diagnostic accuracy of report generation models.

## A.2    IMPLEMENTATION DETAILS

For the abdominal MedVL-CT69K dataset, we reformat all CT scans so that the first axis points from inferior to superior, the second from posterior to anterior, and the third from left to right. We then resample the in-plane axial images to 1mm resolution and the out-of-plane slice thickness to 5mm spacing using trilinear interpolation. We map the Hounsfield unit range -300:400 to the range 0:1, clipping values that fall outside of this range. We use ViT-base Dosovitskiy et al. (2020), initialized with MAE ImageNet-1K pre-trained weights He et al. (2022), as the image encoder. The patch size is set to 16, 16, 32 along the axial, coronal, and sagittal axes, respectively. A pre-trained BERT-base Devlin et al. (2018) model is used as the text encoder. We train fVLM with an Adam optimizer. The learning rate linearly increases to 1e-4 in the first epoch and then decreases to 1e-6 with a cosine decay scheduler. The model undergoes training for 20 epochs on 4 A100 GPUs, with a batch size of 48. During model training, we apply RandomCrop and RandomFlip on the fly. The cropping size is set to 96, 256, and 384 along the axial, coronal, and sagittal axes, respectively. Notably,

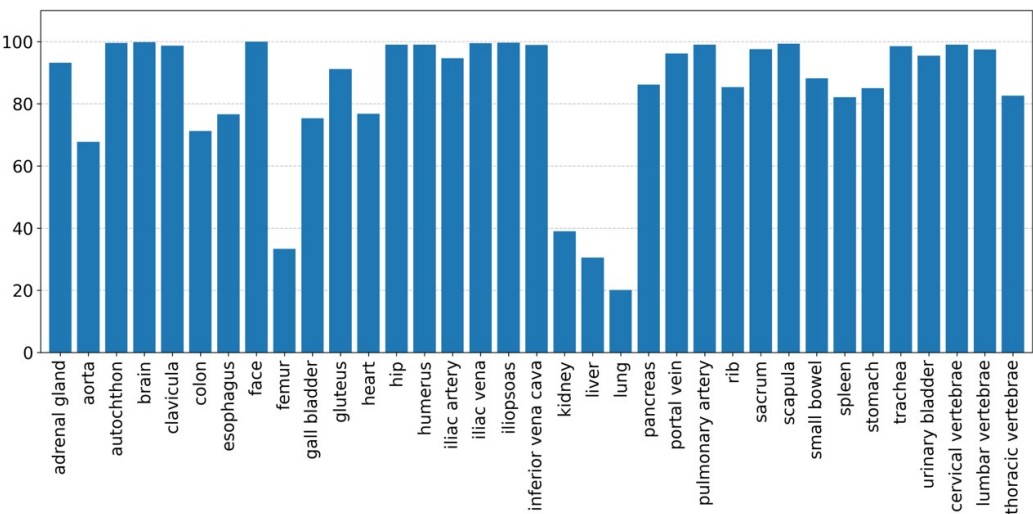

Figure 8: Percentage of normal samples for each anatomy.

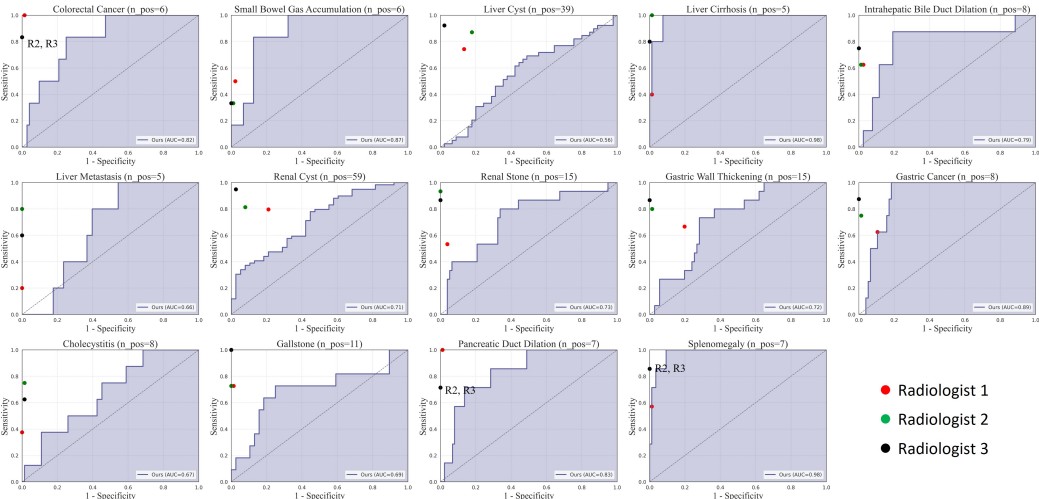

Figure 9: Performance comparison between our method and three radiologists. "n_pos" denotes the number of positive samples of each abnormality.

we observe that if a completely random cropping strategy is used, larger anatomies are more likely to be incomplete after cropping and consequently excluded from the loss calculation. This would introduce a data bias and potentially compromise the model's performance. To address this issue, we employ a uniform sampling strategy to randomly select an anatomy that must be completely included in the cropped image region. For the chest CT-RATE dataset, we apply the same image pre-processing as CT-CLIP Hamamci et al. (2024) to ensure a fair comparison with the competitors. In our co-teaching approach, we iteratively train two fVLMs, alternating between them after each iteration. We initiate a burn-in stage of 5 epochs to allow both models to establish a baseline level of performance. After that, we leverage each model to generate soft labels for its counterpart.

## A.3 READER STUDY

To further validate our method's efficacy, we conduct a reader study to compare our approach with three board-certified radiologists. For this experiment, we randomly select 100 patients from the test set of MedVL-CT69K. Fig. 9 shows the results. Although our method has demonstrated sig-

Table 5: Performance of fVLM when using different models to correct contrastive labels.

|  | Baseline | Model self | Momentum | CoT |
|---|---|---|---|---|
| AUC | 78.7 | 73.3 | 78.8 | 79.8 |
| ACC | 75.3 | 71.4 | 75.5 | 75.9 |

Figure 10: Difference between training model and label correction model.

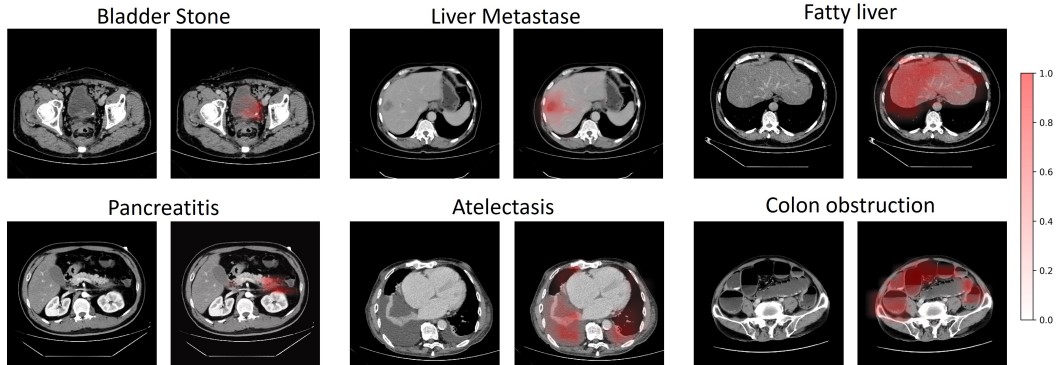

Figure 11: Visual activation maps of our model in diagnosing multiple diseases.

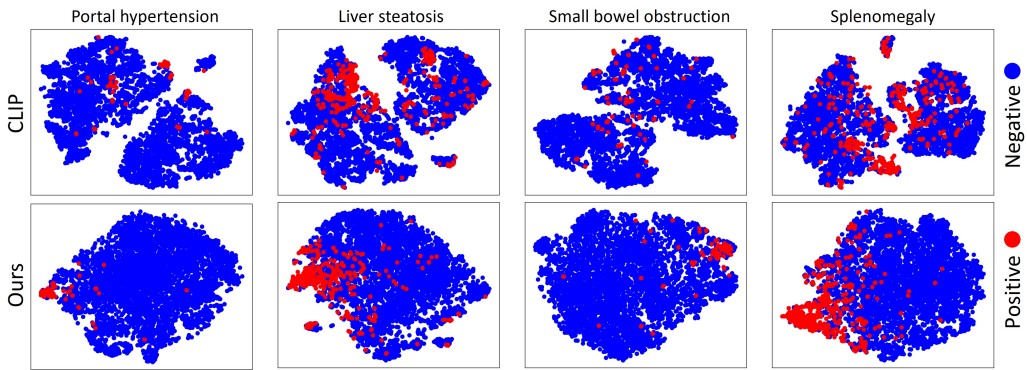

Figure 12: T-SNE visualization of visual embeddings for various abnormalities.

nificant improvements over previous approaches, there remains a noticeable performance gap compared to professional radiologists overall. However, for some diseases such as liver cirrhosis and splenomegaly, our method achieves comparable diagnostic accuracy to radiologists.

## A.4 FURTHER ABLATION ANALYSIS

In Tab. 5, we compare the performance of fVLM when employing different models to correct contrastive learning labels during pre-training. It can be seen that utilizing the training model itself for label correction leads to a significant performance degradation, which could be attributed to the error accumulation issue. Moreover, the proposed CoT strategy yields greater performance gains compared to the momentum model. To explore this, we measure the difference between training model and label correction model by calculating the Euclidean distance of their parameters, as illustrated in Fig. 10. It can be observed that the momentum model, updated through exponential moving average, exhibit minimal discrepancy with the training model. This suggests they may produce similar predictions, potentially leading to error accumulation in the label correction process. In contrast, the iteratively trained models in our proposed CoT framework exhibit considerable distinctness, leading to diverse predictions and reducing the risk of error accumulation.

## A.5 VISUALIZATION

We qualitatively assess the alignment efficacy of our proposed method through visualization in Fig. 11. The heatmaps illustrates the correlation between anatomy-specific visual tokens and the textual embedding of abnormality. We observe high activation in specific affected areas for both localized lesions (*e.g.*, bladder stone) and diffuse abnormalities (*e.g.*, fatty liver). The results demonstrate the model's capacity to precisely localize pathological changes across a spectrum of conditions. Fig. 12 illustrates the distribution of visual embedding for a diverse array of abnormalities. In contrast to CLIP, our method exhibits more compact embedding clusters among positive cases of each abnormality. These findings demonstrate the improved semantic understanding and diagnostic interpretability of our fVLM.

Table 6: Anatomy grouping.

| Anatomical System | Anatomy | Grouped Anatomy |
|---|---|---|
| Organs | Face | Face |
| | Brain | Brain |
| | Esophagus | Esophagus |
| | Trachea | Trachea |
| | Lung upper lobe left
Lung lower lobe left
Lung upper lobe right
Lung middle lobe right
Lung lower lobe right | Lung |
| | Heart myocardium
Heart atrium left
Heart atrium right
Heart ventricle left
Heart ventricle right | Heart |
| | Adrenal gland right
Adrenal gland left | Adrenal gland |
| | Kidney right
Kidney left | Kidney |
| | Stomach | Stomach |
| | Liver | Liver |
| | Gall bladder | Gall bladder |
| | Pancreas | Pancreas |
| | Spleen | Spleen |
| | Colon | Colon |
| | Small bowel
Duodenum | Small bowel |
| | Urinary bladder | Urinary bladder |
| Vessels | Aorta | Aorta |
| | Inferior vena cava | Inferior vena cava |
| | Portal vein and splenic vein | Portal vein and splenic vein |
| | Pulmonary artery | Pulmonary artery |
| | Iliac artery left
Iliac artery right | Iliac artery |
| | Iliac vena left
Iliac vena right | Iliac vena |
| Bones | Vertebrae L1-L4 | Lumbar vertebrae |
| | Vertebrae T1-T12 | Thoracic vertebrae |
| | Vertebrae C1-C7 | Cervical vertebrae |
| | Rib left 1-12
Rib right 1-12 | Rib |
| | Humerus left
Humerus right | Humerus |
| | Scapula left
Scapula right | Scapula |
| | Clavicula left
Clavicula right | Clavicula |
| | Femur left
Femur right | Femur |
| | Hip left
Hip right | Hip |
| | Sacrum | Sacrum |
| Muscles | Gluteus maximus left
Gluteus maximus right
Gluteus medius left
Gluteus medius right
Gluteus minimus left
Gluteus minimus right | Gluteus |
| | Iliopsoas left
Iliopsoas right | Iliopsoas |
| | Autochthon left
Autochthon right | Autochthon |

Table 7: Performance of text classifier.

| Anatomical organ | Abnormality | Precision | Recall | F1-score |
|---|---|---|---|---|
| Adrenal gland | Thickening | 1.00 | 0.97 | 0.99 |
| | Nodule | 1.00 | 0.96 | 0.98 |
| Bladder | Diverticulum | 0.97 | 0.94 | 0.96 |
| | Stones | 1.00 | 1.00 | 1.00 |
| Colon | Gas | 0.84 | 0.79 | 0.81 |
| | Effusion | 0.81 | 0.71 | 0.75 |
| | Obstruction | 0.86 | 1.00 | 0.92 |
| | Diverticulum | 0.97 | 1.00 | 0.98 |
| | Colorectal Cancer | 0.97 | 0.95 | 0.96 |
| | Rectal Cancer | 1.00 | 0.95 | 0.97 |
| | Appendicitis | 1.00 | 1.00 | 1.00 |
| | Appendicolith | 0.89 | 0.96 | 0.92 |
| Esophagus | Hiatal Hernia | 0.74 | 1.00 | 0.85 |
| | Varicose Veins | 1.00 | 1.00 | 1.00 |
| Gallbladder | Cholecystitis | 0.99 | 1.00 | 0.99 |
| | Gallstone | 1.00 | 1.00 | 1.00 |
| | Adenomyomatosis | 0.92 | 0.92 | 0.92 |
| Heart | Cardiomegaly | 1.00 | 0.95 | 0.97 |
| | Pericardial Effusion | 1.00 | 1.00 | 1.00 |
| Kidney | Atrophy | 0.97 | 0.88 | 0.92 |
| | Cyst | 0.97 | 0.96 | 0.97 |
| | Hydronephrosis | 0.88 | 1.00 | 0.94 |
| | Calculi | 0.99 | 0.98 | 0.99 |
| Liver | Steatosis | 0.99 | 1.00 | 0.99 |
| | Glisson's Capsule Effusion | 0.85 | 0.89 | 0.87 |
| | Metastase | 0.90 | 0.95 | 0.92 |
| | Intrahepatic Bile Duct Dilatation | 0.96 | 0.97 | 0.97 |
| | Cancer | 1.00 | 1.00 | 1.00 |
| | Cyst | 0.99 | 0.99 | 0.99 |
| | Abscess | 0.91 | 0.95 | 0.93 |
| | Cirrhosis | 1.00 | 1.00 | 1.00 |
| Lung | Atelectasis | 0.96 | 0.98 | 0.97 |
| | Bronchiectasis | 0.97 | 0.9 | 0.93 |
| | Emphysema | 1.00 | 0.96 | 0.98 |
| | Pneumonia | 0.98 | 0.96 | 0.97 |
| | Pleural effusion | 0.98 | 1.00 | 0.99 |
| Pancreas | Pancreatic cancer | 1.00 | 0.89 | 0.94 |
| | Atrophy | 1.00 | 0.82 | 0.90 |
| | Pancreatitis | 1.00 | 1.00 | 1.00 |
| | Pancreatic duct dilatation | 0.98 | 0.91 | 0.95 |
| | Steatosis | 0.97 | 0.87 | 0.92 |
| Portal vein | Hypertension | 1.00 | 0.91 | 0.95 |
| | Thrombosis | 0.74 | 0.74 | 0.74 |
| Small Intestine | Gas | 0.89 | 0.93 | 0.91 |
| | Effusion | 0.89 | 0.92 | 0.91 |
| | Obstruction | 0.93 | 0.93 | 0.93 |
| | Diverticulum | 0.97 | 1.00 | 0.98 |
| | Intussusception | 0.93 | 0.90 | 0.92 |
| Spleen | Hemangioma | 1.00 | 0.97 | 0.87 |
| | Infarction | 0.95 | 0.97 | 0.96 |
| | Splenomegaly | 1.00 | 0.99 | 1.00 |
| Stomach | Gastric wall thickening | 0.96 | 0.96 | 0.96 |
| | Stomach cancer | 1.00 | 0.97 | 0.99 |
| Sacrum | Osteitis | 0.97 | 1.00 | 0.99 |
| Average | | 0.95 | 0.95 | 0.95 |

Table 8: The distribution of 54 tested abnormalities in the train set. We employ the well-developed text classifier to automatically extract abnormality labels from radiology reports for each sample.

| Anatomy | Anatomy count | Abnormality | Abnormality count |
|---|---|---|---|
| Adrenal gland | 63915 | Thickening | 3037 |
| | | Nodule | 3687 |
| Bladder | 62182 | Diverticulum | 283 |
| | | Stones | 109 |
| Colon | 62054 | Gas | 2173 |
| | | Effusion | 975 |
| | | Obstruction | 436 |
| | | Diverticulum | 1623 |
| | | Colorectal Cancer | 817 |
| | | Rectal Cancer | 858 |
| | | Appendicitis | 1623 |
| | | Appendicolith | 1119 |
| Esophagus | 2636 | Hiatal Hernia | 184 |
| | | Varicose Veins | 609 |
| Gallbladder | 63407 | Cholecystitis | 3935 |
| | | Gallstone | 5500 |
| | | Adenomyomatosis | 1246 |
| Heart | 3701 | Cardiomegaly | 316 |
| | | Pericardial Effusion | 1067 |
| Kidney | 63618 | Atrophy | 921 |
| | | Cyst | 27019 |
| | | Hydronephrosis | 1140 |
| | | Calculi | 5356 |
| Liver | 63690 | Steatosis | 4872 |
| | | Glisson's Capsule Effusion | 915 |
| | | Metastase | 2403 |
| | | Intrahepatic Bile Duct Dilatation | 6093 |
| | | Cancer | 888 |
| | | Cyst | 21710 |
| | | Abscess | 239 |
| | | Cirrhosis | 1772 |
| Lung | 6598 | Atelectasis | 1988 |
| | | Bronchiectasis | 781 |
| | | Emphysema | 190 |
| | | Pneumonia | 1463 |
| | | Pleural effusion | 4665 |
| Pancreas | 63627 | Pancreatic cancer | 933 |
| | | Atrophy | 942 |
| | | Pancreatitis | 1035 |
| | | Pancreatic duct dilatation | 2697 |
| | | Steatosis | 846 |
| Portal vein | 63855 | Hypertension | 1149 |
| | | Thrombosis | 760 |
| Small Intestine | 62419 | Gas | 2906 |
| | | Effusion | 2326 |
| | | Obstruction | 1174 |
| | | Diverticulum | 2352 |
| | | Intussusception | 168 |
| Spleen | 63749 | Hemangioma | 718 |
| | | Infarction | 374 |
| | | Splenomegaly | 1732 |
| Stomach | 63682 | Gastric wall thickening | 2871 |
| | | Stomach cancer | 1064 |
| Sacrum | 62055 | Osteiti | 246 |

Table 9: The distribution of 36 annotated abnormalities in the validation set.

| Anatomy | Anatomy count | Abnormality | Abnormality count |
|---|---|---|---|
| Adrenal gland | 1149 | Thickening | 62 |
| | | Nodule | 79 |
| Colon | 1127 | Gas | 29 |
| | | Effusion | 13 |
| | | Obstruction | 5 |
| | | Colorectal Cancer | 10 |
| | | Rectal Cancer | 16 |
| | | Appendicitis | 5 |
| Gallbladder | 1054 | Cholecystitis | 73 |
| | | Gallstone | 127 |
| | | Adenomyomatosis | 35 |
| Kidney | 1148 | Atrophy | 16 |
| | | Cyst | 492 |
| | | Hydronephrosis | 14 |
| | | Calculi | 104 |
| Liver | 1146 | Steatosis | 97 |
| | | Glisson's Capsule Effusion | 20 |
| | | Metastase | 40 |
| | | Intrahepatic Bile Duct Dilatation | 132 |
| | | Cancer | 10 |
| | | Cyst | 381 |
| | | Cirrhosis | 30 |
| Pancreas | 1149 | Pancreatic cancer | 5 |
| | | Atrophy | 16 |
| | | Pancreatitis | 26 |
| | | Pancreatic duct dilatation | 49 |
| Portal vein | 1150 | Hypertension | 18 |
| | | Thrombosis | 10 |
| Small Intestine | 1131 | Gas | 34 |
| | | Effusion | 25 |
| | | Obstruction | 8 |
| Spleen | 1140 | Hemangioma | 12 |
| | | Infarction | 8 |
| | | Splenomegaly | 35 |
| Stomach | 1150 | Gastric wall thickening | 61 |
| | | Stomach cancer | 20 |

Table 10: The distribution of 54 annotated abnormalities in the test set.

| Anatomy | Anatomy count | Abnormality | Abnormality count |
|---|---|---|---|
| Adrenal gland | 3418 | Thickening | 96 |
| | | Nodule | 87 |
| Bladder | 3243 | Diverticulum | 21 |
| | | Stones | 28 |
| Colon | 3213 | Gas | 129 |
| | | Effusion | 50 |
| | | Obstruction | 17 |
| | | Diverticulum | 104 |
| | | Colorectal Cancer | 96 |
| | | Rectal Cancer | 73 |
| | | Appendicitis | 19 |
| | | Appendicolith | 74 |
| Esophagus | 105 | Hiatal Hernia | 10 |
| | | Varicose Veins | 78 |
| Gallbladder | 3134 | Cholecystitis | 246 |
| | | Gallstone | 355 |
| | | Adenomyomatosis | 60 |
| Heart | 234 | Cardiomegaly | 20 |
| | | Pericardial Effusion | 77 |
| Kidney | 3313 | Atrophy | 37 |
| | | Cyst | 1646 |
| | | Hydronephrosis | 87 |
| | | Calculi | 408 |
| Liver | 3281 | Steatosis | 263 |
| | | Glisson's Capsule Effusion | 68 |
| | | Metastase | 122 |
| | | Intrahepatic Bile Duct Dilatation | 264 |
| | | Cancer | 61 |
| | | Cyst | 1264 |
| | | Abscess | 12 |
| | | Cirrhosis | 188 |
| Lung | 126 | Atelectasis | 70 |
| | | Bronchiectasis | 18 |
| | | Emphysema | 10 |
| | | Pneumonia | 72 |
| | | Pleural effusion | 94 |
| Pancreas | 3328 | Pancreatic cancer | 29 |
| | | Atrophy | 37 |
| | | Pancreatitis | 77 |
| | | Pancreatic duct dilatation | 94 |
| | | Steatosis | 45 |
| Portal vein | 3410 | Hypertension | 54 |
| | | Thrombosis | 55 |
| Small Intestine | 3248 | Gas | 188 |
| | | Effusion | 142 |
| | | Obstruction | 61 |
| | | Diverticulum | 113 |
| | | Intussusception | 10 |
| Spleen | 3352 | Hemangioma | 47 |
| | | Infarction | 22 |
| | | Splenomegaly | 353 |
| Stomach | 3373 | Gastric wall thickening | 206 |
| | | Stomach cancer | 117 |
| Sacrum | 3242 | Osteiti | 17 |

Table 11: Detailed zero-shot performance of our method on each abnormality.

| Anatomy | Abnormality | AUC | ACC | Spec | Sens |
|---------|-------------|-----|-----|------|------|
| Adrenal gland | Thickening | 64.6 | 62.1 | 64.8 | 59.4 |
| | Nodule | 66.8 | 64.8 | 63.0 | 66.7 |
| Bladder | Diverticulum | 85.9 | 77.8 | 74.6 | 81.0 |
| | Stones | 82.0 | 75.7 | 80.0 | 71.4 |
| Colon | Gas Accumulation | 88.7 | 80.8 | 78.6 | 82.9 |
| | Effusion | 87.6 | 80.4 | 78.7 | 82.0 |
| | Obstruction | 99.5 | 98.6 | 97.2 | 100 |
| | Diverticulum | 71.7 | 68.7 | 65.3 | 72.1 |
| | Colorectal Cancer | 72.6 | 64.6 | 65.6 | 63.5 |
| | Rectal Cancer | 85.4 | 77.0 | 82.8 | 71.2 |
| | Appendicitis | 74.9 | 71.9 | 70.2 | 73.7 |
| | Appendicolith | 65.7 | 63.1 | 62.6 | 63.5 |
| Esophagus | Hiatal Hernia | 97.7 | 92.6 | 95.1 | 90 |
| | Varicose Veins | 98.8 | 97.9 | 97.1 | 98.7 |
| Gallbladder | Cholecystitis | 67.3 | 62.7 | 61.1 | 64.2 |
| | Gallstone | 64.6 | 61.8 | 58.3 | 65.4 |
| | Adenomyomatosis | 62.6 | 61.0 | 57.0 | 65.0 |
| Heart | Cardiomegaly | 95.1 | 88.7 | 87.3 | 90.0 |
| | Pericardial Effusion | 76.5 | 74.4 | 64.4 | 84.4 |
| Kidney | Atrophy | 96.0 | 91.5 | 91.1 | 91.9 |
| | Cyst | 68.6 | 63.1 | 64.3 | 61.8 |
| | Hydronephrosis | 75.7 | 69.5 | 78.1 | 60.9 |
| | Calculi | 57.9 | 56.6 | 56.4 | 56.9 |
| Liver | Steatosis | 93.3 | 85.1 | 84.6 | 85.6 |
| | Glisson's Capsule Effusion | 86.5 | 78.9 | 82.7 | 75.0 |
| | Metastase | 78.8 | 71.6 | 70.3 | 73.0 |
| | Intrahepatic Bile Duct Dilatation | 76.8 | 70.8 | 68.2 | 73.5 |
| | Cancer | 84.9 | 79.1 | 77.8 | 80.3 |
| | Cyst | 62.9 | 59.4 | 61.4 | 57.3 |
| | Abscess | 81.8 | 79.5 | 75.7 | 83.3 |
| | Cirrhosis | 94.7 | 88.5 | 87.5 | 89.4 |
| Lung | Atelectasis | 94.8 | 89.4 | 89.4 | 89.3 |
| | Bronchiectasis | 81.6 | 74.1 | 76.0 | 72.2 |
| | Emphysema | 75.0 | 69.3 | 68.6 | 70.0 |
| | Pneumonia | 72.8 | 69.0 | 79.8 | 58.2 |
| | Pleural Effusion | 86.7 | 81.3 | 80.5 | 82.0 |
| Pancreas | Pancreatic Cancer | 87.0 | 79.7 | 80.2 | 79.3 |
| | Atrophy | 86.4 | 77.3 | 76.3 | 78.4 |
| | Pancreatitis | 91.0 | 87.6 | 93.3 | 81.8 |
| | Pancreatic Duct Dilatation | 77.2 | 70.9 | 75.8 | 66.0 |
| | Steatosis | 84.9 | 75.7 | 78.1 | 73.3 |
| Portal vein | Hypertension | 96.8 | 92.2 | 95.5 | 88.9 |
| | Thrombosis | 96.6 | 91.8 | 90.8 | 92.7 |
| Small Intestine | Gas Accumulation | 84.1 | 77.2 | 84.7 | 69.7 |
| | Effusion | 81.5 | 74.2 | 72.4 | 76.1 |
| | Obstruction | 95.2 | 90.5 | 92.6 | 88.5 |
| | Diverticulum | 73.0 | 67.1 | 65.1 | 69.0 |
| | Intussusception | 76.5 | 72.4 | 78.0 | 66.7 |
| Spleen | Hemangioma | 63.8 | 60.6 | 59.5 | 61.7 |
| | Infarction | 89.7 | 86.2 | 90.6 | 81.8 |
| | Splenomegaly | 92.4 | 84.8 | 83.8 | 85.8 |
| Stomach | Gastric Wall Thickening | 69.6 | 65.7 | 62.5 | 68.9 |
| | Gastric Cancer | 78.7 | 72.0 | 73.1 | 70.9 |
| Sacrum | Osteiti | 87.5 | 85.8 | 83.3 | 88.2 |
| Average | | 81.3 | 76.2 | 76.5 | 75.8 |