# OpenReview forum: "Large-scale and Fine-grained Vision-language Pre-training for Enhanced CT Image Understanding"
_ICLR.cc/2025/Conference — ICLR 2025 Spotlight_

### Official Review · Reviewer_5y7k · 2024-10-30

**Soundness:** 4
**Presentation:** 3
**Contribution:** 3
**Rating:** 8
**Confidence:** 5

**Summary:**

The paper presents fVLM, a fine-grained vision-language model for CT image interpretation. The key innovation is to perform anatomy-level contrastive learning between CT scans and radiology reports. The authors introduce a dual false negative reduction approach and curate a large dataset (MedVL-CT69K) with 69,086 patients. The model achieves strong zero-shot performance across 54 disease diagnosis tasks and outperforms existing methods on public benchmarks.

**Strengths:**

1. Anatomy-level fine-grained alignment for the complex report generation task/ diagnosis task is a great idea.
2. The paper collects large-scale datasets with expert annotations, which is time-consuming and expensive in the medical domain.
2. The paper is well-structured, with extensive results (including ablation studies and visualizations) showing clear improvements over baselines.

**Weaknesses:**

The evaluation metrics for the report generation task are limited to basic metrics (BLEU-4) and diagnostic accuracy. The paper should include medical-specific metrics like RadCliQ, RaTESCore, or GREEN that better capture clinical relevance and factual consistency.

**Questions:**

1. Will the MedVL-CT69K dataset or the pre-trained model checkpoint be open-sourced? Additionally, why are there different numbers of diseases annotated in the validation (36) and test (54) sets?
2. While Figure 7 shows more compact embedding clusters for positive cases compared to CLIP, there seems to be limited separation between positive and negative cases.
3. It would be great if the radar plot could link with positive case numbers and normal/abnormal ratios.  This would help readers better understand the model's performance across different conditions and data distributions. Also more analysis on long-tailed disease.
4. The scaling law curve is interesting, 272,124 CT scans from 69,086 unique patients are the largest CT dataset till now, but still far away from data saturation. Do you do the data selection (try to include more abnormal patients) when collecting the dataset? Will the scaling law related to data distribution?

---

> ### Author Response · Authors · 2024-11-20
> **Response to Reviewer 5y7k (1/2)**
>
> Dear 5y7k,
>
> Thank you for your thoughtful feedback and constructive suggestions. We appreciate the opportunity to address your questions and comments, and we provide detailed responses below.
>
> **Q1** The evaluation metrics for the report generation task are limited to basic metrics (BLEU-4) and diagnostic accuracy.
>
> **A1** Thank you for your constructive comment. As suggested, in the revised paper, we have included more metrics (GREEN, BLEU 1-3, METEOR, ROUGE-L, CIDEr) for performance comparison on the downstream report generation task. We present the revised Table 2 as follows:
>
> |   Encoder   |  Init  |     ACC     |    GREEN    |   BLEU-1    |   BLEU-2    |   BLEU-3    |   BLEU-4    |   METEOR    |   ROUGE-L   |    CIDEr    |
> | :---------: | :----: | :---------: | :---------: | :---------: | :---------: | :---------: | :---------: | :---------: | :---------: | :---------: |
> |   Frozen    |   IN   |    51.3     |    34.0     |    47.4     |  32.2   |  25.5   |  21.2   |    28.1     |    44.3     |    10.6     |
> |             |   SP   |    55.8     |    25.9     |    48.3     |    26.3     |    18.0     |    12.8     |    30.8     |    40.6     |     6.6     |
> |             |  MAE   |    50.7     |    21.6     |    49.0     |    27.1     |    18.6     |    13.1     |    30.5     |    41.6     |     6.1     |
> |             |  CLIP  |    57.3     |    33.4     | 49.7 |    28.8     |    20.7     |    15.5     | 31.0 |    42.2     |     9.6     |
> |             |  BIUD  |    58.4     |    33.7     |    47.1     |    30.4     |    23.4     |    18.9     |    29.1     | 44.2 |    13.9     |
> |             | Merlin | 58.8 | 34.2 |    49.5     | 31.4 |    23.9     |    19.0     |    30.0     |    43.8     | 14.3 |
> |             |  Ours  |  61.5   |  37.2   |  50.7   |  32.2   | 24.5 | 19.6 |  31.3   |  45.1   |  14.9   |
> | Fine-tuning |   IN   |    58.0     |    33.4     |    49.4     |    29.8     |    21.9     |    16.9     |    30.4     |    43.6     |    10.0     |
> |             |   SP   |    60.6     |    35.5     |    49.9     |    30.7     |    22.9     |    17.9     |    30.6     |    43.4     |    11.7     |
> |             |  MAE   |    54.4     |    29.4     |    49.0     |    27.1     |    18.6     |    15.1     |    30.5     |    42.5     |     8.8     |
> |             |  CLIP  |    62.0     |    37.6     |    49.9     |    31.9     |    24.3     |    19.5     |    30.7     |    44.7     |    14.3     |
> |             |  BIUD  |    62.6     |    38.8     |    50.2     |    31.7     |    24.0     |    19.0     |    30.9     |    44.7     |    13.8     |
> |             | Merlin | 63.0 | 39.2 | 50.7 |    33.3     | 25.8 | 20.9 | 31.1 | 46.0 |  17.2   |
> |             |  Ours  |  64.5   |  40.2   |  52.2   |  34.5   |  26.8   |  21.9   |  31.6   |  46.4   | 17.1 |
>
> **Q2.1** Will the MedVL-CT69K dataset or the pre-trained model checkpoint be open-sourced?
>
> **A2.1** We sincerely apologize for not clearly claiming the dataset release issue. The MedVL-CT69K dataset is subject to legal matters, and individual participant data will be made available upon request to leading physicians. We are actively working through our internal approval process and will make the code, and model checkpoints public upon approval of our institution, which will take 2 to 3 months.
>
>
> **Q2.2** Different numbers of diseases annotated in the validation (36) and test (54) sets.
>
> **A2.2** Our dataset follows real-world patient distribution, and the validation and test sets are divided from the entire dataset with 1,151 and 3,459 patients, respectively.  Both validation and test sets were annotated by radiologists for 54 disease types. Due to the limited number of patients, the validation set only covers 36 disease types.
>
>
> **Q3** Limited separation between positive and negative cases in Fig. 7.
>
> **A3** Thank you for raising this important point. Our anatomical embeddings are not specific to any one disease but are designed for general diagnostic scenarios. Consequently, the embeddings encompass a rich array of semantic attributes, including the size, shape, texture, density, position, intensity of the organs, as well as characteristics related to the presence of lesions. Even if a sample does not exhibit a specific disease, it may still share certain attributes with some positive samples of that disease, leading to limited separation in the embedding space. For example, a positive sample for liver steatosis and a negative sample may both present liver cysts.

---

> ### Author Response · Authors · 2024-11-20
> **Response to Reviewer 5y7k (2/2)**
>
> **Q4** Model's performance across different conditions and data distributions. More analysis on long-tailed disease.
>
> **A4** Thank you for this insightful question regarding the long-tail distribution of diseases and its impact on model performance. This is an excellent suggestion, and we appreciate the opportunity to expand on this important aspect. In fact, we did not observe a clear relationship between the model's performance for a disease and its frequency of occurrence. More specifically, we calculate the Pearson correlation coefficients between the diagnostic performance (i.e., AUC) and the positive case ratio across all abnormalities on the MedVL-CT69K, CT-RATE, and RadChest-CT datasets. Our method yields results of 0.08, -0.40, and -0.09 and CT-CLIP shows values of 0.06, -0.46, and -0.12. Actually, we observe that the models typically excel at detecting diffuse abnormalities, such as liver cirrhosis and pericardial effusion, while struggling to identify subtle lesions, such as liver cyst and lung nodule. For example, on the CT-RATE benchmark, the positive sample proportions for pericardial effusion and lung nodule are 2.7% and 45.0%, respectively. Despite the much higher positive sample ratio for lung nodule, our method achieves a higher AUC of 0.85 for pericardial effusion diagnosis compared to 0.63 for lung nodule. Similarly, CT-CLIP achieves AUC values of 0.75 and 0.57 for pericardial effusion and lung nodule diagnosis, respectively. We speculate that identifying these subtle lesions relies more on low-level visual features, whereas our current method only leverages the features from the last ViT block. Leveraging hierarchical visual features during training may improve the model's ability in diagnosing these localized lesions.
>
>
>
> **Q5.1** Was data selection applied during dataset collection to include more abnormal patients?
>
> **A5.1** Thank you for your insightful comment. The MedVL-CT69K dataset used in our study was collected continuously over four years, without any specific selection to include more abnormal patients.
>
>
>
> **Q5.2** Will the scaling law related to data distribution?
>
> **A5.2** Yes, the scaling law is relevant to data distribution. Specifically, we analysis the scaling law of our method on the CT-RATE dataset under the same experimental settings as MedVL-CT69K. On these two datasets, we observe different scaling laws: $AUC=0.4557D^{0.0542}$ for CT-RATE and $AUC=0.427D^{0.0598}$ for MedVL-CT69K. The scaling law for CT-RATE exhibits a larger coefficient but a smaller exponent. This indicates that in the low-data regime, our model tends to perform better on the CT-RATE dataset, while as data volume increases, the performance improvement on CT-RATE becomes less pronounced, reaching saturation more quickly. This can be attributed to the relatively lower complexity of the chest CT images in the CT-RATE dataset, which involve fewer anatomical structures and less pathological diversity compared to the abdominal CT images in the MedVL-CT69K dataset.
>
> ***
> We would like to sincerely thank you for your thoughtful and constructive feedback during the initial review process. Your comments have been instrumental in improving the quality and clarity of our work. We have carefully addressed all of your concerns to the best of our ability and hope that our responses and revisions adequately resolve the issues you raised.

---

> > ### Comment · Reviewer_5y7k · 2024-11-20
> >
> > Thank you for your detailed responses and analysis.  I will increase my score correspondingly.

---

### Official Review · Reviewer_wu1u · 2024-11-02

**Soundness:** 3
**Presentation:** 2
**Contribution:** 3
**Rating:** 8
**Confidence:** 4

**Summary:**

The authors present an vision-language model for  anatomy-aware medical report generation, adressing the task of false negative reduction.  A large (private) dataset is curated and good performances on public datasets are presented.

**Strengths:**

* The study addresses an interesting multi-modal learning problem from the medical domain
* The presented false negative reduction is a meaningful integration to vision-language model report generation
* The results are promising indicating the relevance of this addition

**Weaknesses:**

* The methodology is not quite clear
* There is no codebase, preproduction may be a challenge.

**Questions:**

* Image Encoder Setup: The authors mention using a ViT base model with MAE pretraining on ImageNet-1K and a patch size of 16 x 16 x 32 (referenced in A.2). However, it’s unclear how they process 3D volumes with 3D patch embeddings through a 2D ViT model. Although they provide details on the training procedure, the overall model architecture remains vague. Given the lack of source code, a clearer description of the model's structure is crucial. Can the authors improve?
* Comparison with 2D CLIP Methods: The comparisons made with 2D CLIP methods in Table 1 are not fully explained. Specifically, it's unclear how they adapted these methods to a 3D context and how the training was conducted. Please clarify.
* Evaluation with CT-CLIP: In Table 3, the comparison with CT-CLIP models is ambiguous. Did the authors fine-tune a pre-trained model (based on their in-house data) on CT-RATE, or was there a different approach taken?
* t-SNE Consistency: To enhance the consistency of the embedding spaces shown in Fig. 7, the authors could consider setting a random seed before calculating the t-SNE embeddings.
* HU Value Clipping: The authors chose to clip HU values between -300 and 400, which raises a question: Does this range cover all abnormalities? For instance, lung-related abnormalities are typically closer to -1000 (representing air), while bone-related abnormalities can reach values close to 1000. Since they utilized both non-contrast and contrast-enhanced CT scans (arterial, venous, and delayed phases), did they apply this same HU preprocessing across all scans? Additionally, did the authors use this range for the CT-RATE training ablations (which is different than the original CT-RATE paper)? Please clarify.

---

> ### Author Response · Authors · 2024-11-20
> **Response to Reviewer wu1u**
>
> Dear wu1u,
>
> Thank you for your detailed and insightful feedback. We appreciate your recognition of the strengths of our approach, and your constructive suggestions highlight important areas for further improvement. Below, we address each of your comments and questions:
>
>
>
> **Q1** How to process 3D volumes with a 2D ViT model.
>
> **A1** Thank you for the thoughtful feedback. After patchifying and flattening the input, both 2D and 3D images are converted into 1D token sequences, allowing us to process them using the same ViT model. The pre-trained ViT model includes four main components: a 2D convolution layer for patch embedding, a [cls] token, a learnable positional embedding layer, and a stack of Transformer blocks. To adapt the pre-trained ViT model for volume input, we replace the 2D convolution layer with a randomly initialized 3D convolution layer to perform patch embedding. For the positional embeddings, we first interpolate them across the spatial dimensions (i.e., coronal and sagittal axes) and then replicate them along the axial axis to match the size of patchified volumes. All other parameters are directly reused from the pretrained model.
>
>
>
> **Q2** It's unclear how the 2D CLIP methods are adapted to a 3D context and how the training was conducted.
>
> **A2** To adapt the 2D CLIP methods for 3D context, we replace their original 2D image encoders with the same image encoder used in our method. This modification enables these methods to process 3D volumetric inputs without altering their core innovations. Moreover, the text encoder is also replaced with the same one used in our method to ensure a fair comparison. All methods are then trained and evaluated under the same experimental setup.
>
>
>
> **Q3**  In Table 3, the comparison with CT-CLIP models is ambiguous. Did the authors fine-tune a pre-trained model (based on their in-house data) on CT-RATE?
>
> **A3** To ensure a fair comparison with CT-CLIP models, we do not initialize our model with the pre-trained weights on the large-scale in-house data. Following the official settings used in CT-CLIP, we train our model on the CT-RATE training set from scratch and use the CT-RATE test set for internal validation and the whole Rad-ChestCT dataset for external evaluation.
>
>
>
> **Q4**  To enhance the consistency of the embedding spaces shown in Fig. 7, the authors could consider setting a random seed before calculating the t-SNE embeddings.
>
> **A4** Thanks for your constructive comment. In fact, we have set a fixed seed before calculating the t-SNE embeddings. We would like to clarify that each feature point in Fig. 7 represents an anatomy-level feature, with each row of four figures corresponding to the features of the portal vein, liver, small bowel, and spleen, respectively. As a result, the embedding distributions vary naturally across these anatomical structures.
>
> **Q5** The authors clipped HU values to -300 to 400, raising questions about whether this range covers all abnormalities. Was this preprocessing applied across both non-contrast and contrast-enhanced scans? Additionally, was this range used for CT-RATE training, differing from the original CT-RATE paper?
>
> **A5** Thank you for your thoughtful and detailed feedback. The MedVL-CT69K dataset is derived from abdominal CT scans, and the primary goal is to diagnose abdominal diseases. In this context, we apply the soft tissue window for image normalization. In fact, we experimented with different HU clipping ranges and found that the range of [-300, 400] yielded better results compared to the broader range of [-1000, 1000]. This is likely because most of the 54 diseases we evaluated can be effectively diagnosed within the soft tissue window in both non-contrast and contrast-enhanced CT scans, and the narrower range improves image contrast, enabling the model to better capture features relevant to these diseases. In future work when training our model for whole-body diagnosis tasks, we will consider wider HU ranges to account for the lung-related abnormalities with HU values closer to -1000 and bone-related abnormalities that reach values close to 1000. For the chest CT-RATE dataset, we apply the same image pre-processing steps as CT-CLIP to ensure a fair comparison with the competitors. Specifically, we clip the HU values to a range of -1000 to 1000 and normalize them to range [-1, 1] during model training. We have clarified this point in the revised paper, as detailed in the Implementation Details section in Appendix A.2. Thanks again for your constructive comment.
>
> ***
> We express our sincere gratitude for your valuable feedback during the initial review, especially for your careful attention to detail. Thank you for providing us an opportunity to clarify the points you raised. The insights you provide are indispensable to us, enabling significant improvements in the quality of our work. We are hopeful that our responses adequately address your concerns.

---

> > ### Comment · Reviewer_wu1u · 2024-11-24
> > **codebase?**
> >
> > Thank you for your response. I might have missed it, but can you comment on your plans to make the code / models available?

---

> ### Author Response · Authors · 2024-11-25
>
> Dear Reviewer wu1u,
>
> We have provided the code through an anonymous link: https://anonymous.4open.science/r/fVLP-DF95 to facilitate the reviewing process. Besides, we are actively working through our internal approval process, which typically requires 2 to 3 months, and will make the code and model checkpoints public upon approval of our institution.

---

> > ### Comment · Area_Chair_iRJK · 2024-11-25
> >
> > Dear Authors,
> >
> > Could you please provide an estimation of when your institute is likely to approve the publication of the code? Please note that the promise of releasing the code will be included as part of the public record for this paper.

---

> ### Author Response · Authors · 2024-11-25
>
> Dear Area Chair iRJK,
>
> Thank you for your inquiry regarding the release of the code. The approval process within our institute takes 2 to 3 months. We are closely following the approval procedure, and the code will be released immediately upon approval.

---

> > ### Comment · Area_Chair_iRJK · 2024-11-25
> >
> > Dear Authors,
> >
> > To assist in the review process, could you confirm whether reviewers can proceed under the presumption that the code will eventually be released, even if it may take 2 to 3 months for your institute to approve the publication? This clarification will help ensure a fair evaluation of your submission.
> >
> > Thank you for your prompt response.

---

### Official Review · Reviewer_oBhB · 2024-11-02

**Soundness:** 3
**Presentation:** 3
**Contribution:** 3
**Rating:** 6
**Confidence:** 4

**Summary:**

This work discusses the fine-grained alignment issue for vision-language clip-like pretraining for CT and reports. To improve the granularity of alignment the authors propose to pre-segment key anatomical structures in CT using a public segmentation model (Totalsegmenter), and to pre-segment descriptions corresponding to individual anatomical structures in the free-text report using an open-access LLM (Qwen). This allows fine-grained contrastive learning at the anatomical structure level. Considerations are made to avoid pushing representations of the same organ/condition away. Improved performances are shown for zero-shot abnormality detection compared with several existing approaches.

**Strengths:**

The idea of increasing the granularity of contrastive learning by data pre-segmentation using public tools is neat, simple, and straightforward.

The paper is written with sufficient clarity, where the insights behind each design and the implementations details are well-presented.

Improved performance on zero-shot abnormality detection is shown compared with some existing works.

**Weaknesses:**

For a comprehensive assessment of report generation some essential scores are missing, such as BLEU 1-3, ROUGE-L, and METEOR. Also, comparisons with vision encoders from peer vision-language pre-training for CT works may be needed. This is my major concern as the authors have claimed improved performance on report generation task.

Despite improved granularity, attributing sentences of reports `mentioning` a structure to the image feature of that structure may sometimes be misleading especially when manual verification/correction is not accessible. E.g., in practice pancreatitis is often associated with the entire abdomen instead of the pancreas alone and neighboring structures. The authors are encouraged to comment on this.

**Questions:**

A high-level question: The capability of open-source image segmentation tools (totalsegmenter and/or SAM family) and open-source LLM advance fiercely -- does it imply that the utility of (un-/weakly-supervised) vision-language pretraining for CT will gradually phase out? One may directly construct semantic labels from text reports using strong enough LLM tools and turn the problem back to large scale supervised learning. I am curious to hear the authors’ insight.

**Details Of Ethics Concerns:**

A separate ICLR ethics reviewer might not be needed. However, as the proposed study involves curating and analyzing human CT images and reports, the authors need to make sure that essential ethical approval is obtained, and pertinent privacy laws/regulations are complied with.

---

> ### Author Response · Authors · 2024-11-20
> **Response to Reviewer oBhB**
>
> Dear Reviewer oBhB,
>
> Thank you for your feedback and constructive suggestions. We appreciate the opportunity to clarify and expand on several aspects of our work. Below, we address each of your points in detail:
>
> **Q1** The evaluation metrics for the report generation task are limited to basic metrics (BLEU-4) and diagnostic accuracy in Table 2.
>
> **A1** Thank you for your constructive comment. As suggested, in the revised paper, we have included more metrics (GREEN, BLEU 1-3, METEOR, ROUGE-L, CIDEr) on the downstream report generation task. We present the revised Table 2 as follows:
>
> |   Encoder   |  Init  | ACC  | GREEN | BLEU-1 | BLEU-2 | BLEU-3 | BLEU-4 | METEOR | ROUGE-L | CIDEr |
> | :---------: | :----: | :--: | :---: | :----: | :----: | :----: | :----: | :----: | :-----: | :---: |
> |   Frozen    |   IN   | 51.3 | 34.0  |  47.4  |  32.2  |  25.5  |  21.2  |  28.1  |  44.3   | 10.6  |
> |             |   SP   | 55.8 | 25.9  |  48.3  |  26.3  |  18.0  |  12.8  |  30.8  |  40.6   |  6.6  |
> |             |  MAE   | 50.7 | 21.6  |  49.0  |  27.1  |  18.6  |  13.1  |  30.5  |  41.6   |  6.1  |
> |             |  CLIP  | 57.3 | 33.4  |  49.7  |  28.8  |  20.7  |  15.5  |  31.0  |  42.2   |  9.6  |
> |             |  BIUD  | 58.4 | 33.7  |  47.1  |  30.4  |  23.4  |  18.9  |  29.1  |  44.2   | 13.9  |
> |             | Merlin | 58.8 | 34.2  |  49.5  |  31.4  |  23.9  |  19.0  |  30.0  |  43.8   | 14.3  |
> |             |  Ours  | 61.5 | 37.2  |  50.7  |  32.2  |  24.5  |  19.6  |  31.3  |  45.1   | 14.9  |
> | Fine-tuning |   IN   | 58.0 | 33.4  |  49.4  |  29.8  |  21.9  |  16.9  |  30.4  |  43.6   | 10.0  |
> |             |   SP   | 60.6 | 35.5  |  49.9  |  30.7  |  22.9  |  17.9  |  30.6  |  43.4   | 11.7  |
> |             |  MAE   | 54.4 | 29.4  |  49.0  |  27.1  |  18.6  |  15.1  |  30.5  |  42.5   |  8.8  |
> |             |  CLIP  | 62.0 | 37.6  |  49.9  |  31.9  |  24.3  |  19.5  |  30.7  |  44.7   | 14.3  |
> |             |  BIUD  | 62.6 | 38.8  |  50.2  |  31.7  |  24.0  |  19.0  |  30.9  |  44.7   | 13.8  |
> |             | Merlin | 63.0 | 39.2  |  50.7  |  33.3  |  25.8  |  20.9  |  31.1  |  46.0   | 17.2  |
> |             |  Ours  | 64.5 | 40.2  |  52.2  |  34.5  |  26.8  |  21.9  |  31.6  |  46.4   | 17.1  |
>
> **Q2**  Attributing sentences of reports mentioning a structure to the image feature of that structure may sometimes be misleading especially when manual verification/correction is not accessible.
>
> **A2** We fully appreciate the reviewer's concern. In this work, we use the Vision Transformer as the image enocder, where the self-attention mechanism enables each anatomical structure to dynamically interact with other structures and extract diagnostic-related cues from them. This results in a context-aware representation of each anatomy, facilitating the diagnosis of diseases that may involve more extensive anatomical areas.
>
> **Q3** The rapid advancement of open-source segmentation tools and LLMs raises the question of whether vision-language pretraining for CT willgradually phase out, as semantic labels could be directly constructed from text reports for supervised learning.
>
> **A3** We appreciate the insightful question and agree that advancements in open-source image segmentation tools and LLMs have expanded the possibilities for constructing semantic labels from text reports, potentially facilitating large-scale supervised learning. However, we believe that supervised learning is not an alternative to vision-language pretraining (VLP). While LLMs can be used to extract semantic labels from reports, this process typically discards many detailed descriptions that are crucial for understanding complex medical conditions. Specifically, these semantic labels are typically encoded as one-hot, which are effective for capturing discrete attributes such as the presence or absence of a disease but fall short in representing more detailed, continuous disease attributes such as size, location and density. These finer details provide essential insights into disease severity, progression, and response to treatment. Moreover, the semantic labels extracted by LLMs are not guaranteed to be 100% accurate, and manual calibration requires significant time and effort, which is not conducive to scaling datasets. In contrast, using diagnostic reports directly as supervision signals in VLP requires no additional annotations, providing a clear advantage in scaling data. Furthermore, reports are condensed records of the entire clinical diagnostic process, capturing detailed diagnostic descriptions and offering strong interpretability with comprehensive information. This enables VLP models to achieve a more holistic and nuanced understanding of complex diseases and their underlying patterns.
>
> ***
> We express our sincere gratitude for your constructive feedback in the initial review. Your expert insights are invaluable to us in our pursuit of elevating the quality of our work.

---

> > ### Comment · Reviewer_oBhB · 2024-11-24
> >
> > I have read through the rebuttal and I would like to thank the authors for the supplemented quantitive results for the report generation tasks and for the insights on the future of vision-language pretraining for radiological data. I would like to remain my positive rating.

---

### Official Review · Reviewer_PfQv · 2024-11-04

**Soundness:** 3
**Presentation:** 3
**Contribution:** 2
**Rating:** 8
**Confidence:** 3

**Summary:**

The paper proposes a fine-grained vision-language model (fVLM) to improve CT image interpretation by associating specific anatomical regions in CT scans with corresponding descriptions in radiology reports, addressing limitations of global image-report contrastive methods. It introduces a large dataset, MedVL-CT69K, encompassing 272,124 CT scans and achieves state-of-the-art performance. Key contributions include anatomy-level contrastive learning and a method to handle false negatives.

**Strengths:**

1. Fine-Grained Anatomy-Level Alignment: The paper introduces a novel, fine-grained vision-language model (fVLM) that aligns anatomical regions in CT images with corresponding report descriptions.
2. Large and Comprehensive Dataset: Experiments were conducted on a range of datasets including the largest CT dataset to date (MedVL-CT69K)
3. Effectively tackling shortcomings of their method by introducing the dual false-negative reduction approach.

**Weaknesses:**

1. Experiments are incomplete: Table 1 doesn't include a performance evaluation of the methods from Table 3, namely CT-CLIP, CT-VocabFine, CT-LiPro. Table 3 doesn't include performance evaluation of the methods from Table 1. In both cases it's not argued why those experiments were not conducted. For Table 1, and Table 2 it's also unclear how the 2D approaches were adapted for 3D and how their evaluation was then done. Additionally, the evaluation metrics in Table 1 and Table 3 differ. Why? Furthermore, the performance comparison in Table 2 should also include comparisons to other CT-CLIP approaches, not only natural image counterparts (he it's again not clear how the 2D models were adapted to 3D).
2. Report generation uses only a single NLU metric. Other NLU metrics such as ROUGE would be interesting. Furthermore, using the GREEN metric[1] or RadFact[2] for evaluation would allow assessing clinical relevance.
3. T-SNE visualization: An (additional) comparison of the embeddings to CT-CLIP would be be interesting to see if they have the same clustering behavior.
4. Ablations study on masking missing: The masking approach is not well investigate and no ablation studies are done. One intuitive baseline would be the anatomical cropping of the image. What's the advantage of the masking approach?
5. The authors approach only works for CT scans, since it relies on anatomy segmentations from TotalSegmentator. Furthermore, only on the classes which are available in TotalSegmentator.

[1] Ostmeier, S., Xu, J., Chen, Z., Varma, M., Blankemeier, L., Bluethgen, C., ... & Delbrouck, J. B. (2024). GREEN: Generative Radiology Report Evaluation and Error Notation. arXiv preprint arXiv:2405.03595.

[2] Bannur, S., Bouzid, K., Castro, D. C., Schwaighofer, A., Bond-Taylor, S., Ilse, M., ... & Hyland, S. L. (2024). MAIRA-2: Grounded Radiology Report Generation. arXiv preprint arXiv:2406.04449.

**Questions:**

1. What's the authors reproducibility statement? Will the dataset, code and weights be released? Especially publishing the dataset would be of great value for the community.

---

> ### Author Response · Authors · 2024-11-20
> **Response to Reviewer PfQv (1/3)**
>
> Dear Reviewer PfQv,
>
> Thank you for your detailed and constructive feedback on our work. We appreciate your insights into both the strengths and weaknesses of our approach, and we address each of your points below:
>
> **Q1.1** Incosistency of compartive methods and metrics between Table. 1 and Table. 3
>
> **A1.1** Thanks for your valuable comment. First, we would like to clarify a misunderstanding: CT-CLIP is a renaming of CLIP in the paper by Hamamci et al. (2024) [1], representing a variant of CLIP applied to the CT scenario, and it denotes the same method as CLIP in our paper. We have clarified this point in the revised paper. Moreover, as suggested, we have included CT-VocabFine and CT-LiPro as supervised fine-tuning methods in Table 1. In Table 3, we have added comparisons with two methods, including BIUD and Merlin, to compare with more state-of-the-art (SOTA) methods. Regarding evaluation metrics, we have included weighted F1 and Precision in Table 1, and we have added Sensitivity (Sens) and Specificity (Spec) in Table 3.
>
> - The revised Table 1
>
> |            | Method       | AUC  | ACC  | Spec | Sens | F1   | Prec |
> | :--------: | ------------ | ---- | ---- | ---- | ---- | ---- | ---- |
> | Supervised | Baseline     | 73.3 | 69.1 | 76.2 | 62.0 | 79.4 | 17.6 |
> |            | CT-VocabFine | 76.7 | 72.2 | 76.1 | 68.2 | 81.6 | 20.3 |
> |            | CT-LiPro     | 76.5 | 70.9 | 76.8 | 65.1 | 81.3 | 19.3 |
> | Zero-shot  | CLIP         | 68.4 | 66.7 | 68.0 | 65.5 | 76.0 | 18.0 |
> |            | LOVT         | 69.4 | 65.4 | 60.1 | 70.8 | 70.9 | 15.2 |
> |            | MGCA         | 70.1 | 66.4 | 64.5 | 68.3 | 73.9 | 16.0 |
> |            | Imitate      | 70.6 | 67.9 | 66.6 | 69.2 | 75.4 | 17.9 |
> |            | ASG          | 70.1 | 67.7 | 67.4 | 68.0 | 75.9 | 18.8 |
> |            | CT-GLIP      | 69.3 | 66.9 | 63.1 | 70.6 | 74.2 | 18.1 |
> |            | BIUD         | 71.4 | 69.2 | 69.0 | 69.3 | 76.6 | 18.7 |
> |            | Merlin       | 71.9 | 69.5 | 69.7 | 69.2 | 77.0 | 18.1 |
> |            | Ours         | 81.3 | 76.2 | 76.5 | 75.8 | 82.2 | 21.1 |
>
> - The revised Table 3
>
> |              Dataset              | Supervised |          |              |          | Zero-shot |      |        |      |
> | :-------------------------------: | :--------: | :------: | :----------: | :------: | :-------: | :--: | :----: | :--: |
> |                                   |   Metric   | Baseline | CT-VocabFine | CT-LiPro |  CT-CLIP  | BIUD | Merlin | Ours |
> |   Internal validation (CT-RATE)   |    AUC     |   60.3   |     75.0     |   75.1   |   70.4    | 71.3 |  72.8  | 77.8 |
> |                                   |    ACC     |   58.1   |     69.2     |   67.6   |   65.1    | 68.1 |  67.2  | 71.8 |
> |                                   |  F1  |   63.2   |     72.8     |   71.4   |   69.1    | 71.6 |  70.9  | 75.1 |
> |                                   | Prec  |   24.0   |     34.2     |   33.1   |   30.6    | 33.8 |  33.7  | 37.9 |
> |                                   |    Spec    |    -     |      -       |    -     |     -     | 68.6 |  66.8  | 71.7 |
> |                                   |    Sens    |    -     |      -       |    -     |     -     | 67.3 |  70.1  | 72.8 |
> | External validation (Rad-ChestCT) |    AUC     |   54.1   |     64.9     |   64.7   |   63.2    | 62.9 |  64.4  | 68.0 |
> |                                   |    ACC     |   53.9   |     62.2     |   62.5   |   59.9    | 60.6 |  61.9  | 64.7 |
> |                                   |  F1  |   58.7   |     66.5     |   66.8   |   64.8    | 65.2 |  66.3  | 68.8 |
> |                                   |    Prec    |   28.7   |     35.9     |   35.3   |   33.9    | 33.7 |  34.8  | 37.4 |
> |                                   |    Spec    |    -     |      -       |    -     |     -     | 60.2 |  61.7  | 64.6 |
> |                                   |    Sens    |    -     |      -       |    -     |     -     | 59.6 |  61.0  | 64.6 |
>
>
> **Q1.2** How to adpat 2D approaches for 3D and how their evaluation was done.
>
> **A1.2** To adapt the 2D approaches for 3D context, we replace their original 2D image encoders with the same image encoder used in our method. This modification enables these competitors to process 3D volumetric inputs without altering their core innovations. Then, we train and evaluate all methods using the same experimental setup.

---

> ### Author Response · Authors · 2024-11-20
> **Response to Reviewer PfQv (2/3)**
>
> **Q2** Missing comparisons with other CT-CLIP approaches in Table 2. It's not clear how the 2D models were adapted to 3D. Report generation uses only a single NLU metric.
>
> **A2** Thank you for the constructive comment. Except IN, all other models in Table 2 are pre-trained on the MedVL-CT69K dataset. As we have adapted these 2D CLIP approaches for 3D inputs in vision-language pre-training, we can easily integrate their pre-trained image encoders with a BERT-base text decoder for CT report generation. In the revised paper, we have included more vision-language pre-training approaches (BIUD, Merlin) and metrics (GREEN [2], BLEU 1-3, METEOR, ROUGE-L, CIDEr) for performance comparison on the downstream report generation task. We present the revised Table 2 as follows:
>
> - The revised Table 2
>
> |   Encoder   |  Init  | ACC  | GREEN | BLEU-1 | BLEU-2 | BLEU-3 | BLEU-4 | METEOR | ROUGE-L | CIDEr |
> | :---------: | :----: | :--: | :---: | :----: | :----: | :----: | :----: | :----: | :-----: | :---: |
> |   Frozen    |   IN   | 51.3 | 34.0  |  47.4  |  32.2  |  25.5  |  21.2  |  28.1  |  44.3   | 10.6  |
> |             |   SP   | 55.8 | 25.9  |  48.3  |  26.3  |  18.0  |  12.8  |  30.8  |  40.6   |  6.6  |
> |             |  MAE   | 50.7 | 21.6  |  49.0  |  27.1  |  18.6  |  13.1  |  30.5  |  41.6   |  6.1  |
> |             |  CLIP  | 57.3 | 33.4  |  49.7  |  28.8  |  20.7  |  15.5  |  31.0  |  42.2   |  9.6  |
> |             |  BIUD  | 58.4 | 33.7  |  47.1  |  30.4  |  23.4  |  18.9  |  29.1  |  44.2   | 13.9  |
> |             | Merlin | 58.8 | 34.2  |  49.5  |  31.4  |  23.9  |  19.0  |  30.0  |  43.8   | 14.3  |
> |             |  Ours  | 61.5 | 37.2  |  50.7  |  32.2  |  24.5  |  19.6  |  31.3  |  45.1   | 14.9  |
> | Fine-tuning |   IN   | 58.0 | 33.4  |  49.4  |  29.8  |  21.9  |  16.9  |  30.4  |  43.6   | 10.0  |
> |             |   SP   | 60.6 | 35.5  |  49.9  |  30.7  |  22.9  |  17.9  |  30.6  |  43.4   | 11.7  |
> |             |  MAE   | 54.4 | 29.4  |  49.0  |  27.1  |  18.6  |  15.1  |  30.5  |  42.5   |  8.8  |
> |             |  CLIP  | 62.0 | 37.6  |  49.9  |  31.9  |  24.3  |  19.5  |  30.7  |  44.7   | 14.3  |
> |             |  BIUD  | 62.6 | 38.8  |  50.2  |  31.7  |  24.0  |  19.0  |  30.9  |  44.7   | 13.8  |
> |             | Merlin | 63.0 | 39.2  |  50.7  |  33.3  |  25.8  |  20.9  |  31.1  |  46.0   | 17.2  |
> |             |  Ours  | 64.5 | 40.2  |  52.2  |  34.5  |  26.8  |  21.9  |  31.6  |  46.4   | 17.1  |
>
>
>
> **Q3** Comparison to the embeddings of CT-CLIP.
>
> **A3** Thank you for the insightful comment. CLIP and CT-CLIP indeed refer to the same method. As such, in our t-SNE visualizations, the embedding distributions from CLIP indeed represent the results of CT-CLIP.
>
>
> **Q5** The proposed approach only works for CT scans and the classes available in TotalSegmentator due to its reliance on anatomy segmentations from this tool.
>
> **A5** Thanks for your thoughtful feedback. While our experiments primarily focus on CT data, the proposed method is inherently general and can be extended to other imaging modalities and anatomical classes, provided that the anatomy masks are available. Indeed, as Reviewer oBhB mentioned, recent years have witnessed significant advancements in universal medical segmentation models, which are now capable of generating a diverse range of high-quality anatomical masks across multiple imaging modalities. For instance, SAT [3] can segment 497 anatomical classes across three modalities (CT, MRI, and PET) and 8 regions of the human body, such as brain, thorax, abdomen, and pelvis. These models greatly facilitate the deployment of our method.
>
>
>
> **Q6** Will the dataset, code and weights be released?
>
> **A6** We sincerely apologize for not clearly claiming the dataset release issue. The MedVL-CT69K dataset is subject to legal matters, and individual participant data will be made available upon request to leading physicians. We have provided the code through an anonymous link: https://anonymous.4open.science/r/fVLP-DF95 to facilitate the reviewing process. Besides, we are actively working through our internal approval process, which typically requires 2 to 3 months, and will make the code and model checkpoints public upon approval of our institution.
>
> ***
> **Q4** Ablation study on masking missing
>
> We would like to kindly ask the reviewer to clarify the question regarding the "masking approach," as our method does not incorporate this module. Furthermore, we also wish to note that the terms "masking" and "missing" do not appear throughout our paper.

---

> > ### Comment · Reviewer_PfQv · 2024-11-25
> >
> > reg. Q4: There is no ablation study examining how the segmentation masks are utilized for foreground and background separation to guide the model in creating anatomy-specific representations (see L260–L265). An intuitive alternative approach could involve cropping the anatomies. It would be interesting and valuable to include ablation studies exploring this direction.

---

> > > ### Comment · Reviewer_PfQv · 2024-11-25
> > >
> > > Thank you for the comprehensive and detailed responses. You have adequately addressed my concerns. I will raise my score.

---

> > > ### Author Response · Authors · 2024-11-27
> > >
> > > Dear Reviewer PfQv,
> > >
> > > Thank you for your insightful question. While cropping anatomies from raw CT images to create anatomy-specific representations seems more intuitive, this strategy inevitably discards contextual information that is crucial for diagnosing certain diseases. For example, as noted by Reviewer oBhB, in practice pancreatitis is often associated with the entire abdomen instead of the pancreas alone and neighboring structures. Similarly, liver-related conditions such as cirrhosis or metastatic lesions often require examining the liver in conjunction with the surrounding vasculature, spleen, and other abdominal anatomies to provide accurate interpretations. To address these limitations, we construct global image representations first and then separate out organ-specific tokens in this work, enabling each anatomy to dynamically interact with other anatomical structures and extract diagnostic-related cues from them. This strategy results in a context-aware representation of each anatomy, facilitating the diagnosis of diseases that may involve more extensive anatomical areas. In the following table, we ablate the model performance with these two strategies on the MedVL-CT69K validation set. It can be seen that our approach yields better results.
> > >
> > > |            Method             | AUC  | ACC  |
> > > | :---------------------------: | :--: | :--: |
> > > | Anatomical cropping of images | 73.3 | 71.4 |
> > > |             Ours              | 79.8 | 75.9 |
> > >
> > > We greatly appreciate your insightful comment. Thank you once again for your time and valuable suggestions.

---

> ### Author Response · Authors · 2024-11-20
> **Response to Reviewer PfQv (3/3)**
>
> Refs:
>
> [1] Hamamci, Ibrahim Ethem, et al. "A foundation model utilizing chest CT volumes and radiology reports for supervised-level zero-shot detection of abnormalities." arXiv preprint arXiv:2403.17834 (2024).
>
> [2] Ostmeier, S., Xu, J., Chen, Z., Varma, M., Blankemeier, L., Bluethgen, C., ... & Delbrouck, J. B. (2024). GREEN: Generative Radiology Report Evaluation and Error Notation. arXiv preprint arXiv:2405.03595.
>
> [3] Zhao, Ziheng, et al. "One model to rule them all: Towards universal segmentation for medical images with text prompts." arXiv preprint arXiv:2312.17183 (2023).
>
> ***
>
> We greatly appreciate your consideration of these clarifications. Your positive comments in the initial review were very encouraging, and we are truly grateful for your thoughtful feedback. We would be extremely appreciative if you could consider further raising the score.
>
> If you still need any clarification or have any other concerns, please do not hesitate to reach out to us for any reason. Your guidance is greatly appreciated, and we are fully committed to a constructive and productive discussion.

---

### Comment · Area_Chair_iRJK · 2024-11-24

Dear Reviewers,

The discussion with the authors will conclude soon. The authors have provided detailed rebuttals. If there are any points that you feel have not been adequately clarified or if there are misunderstandings in their responses, please take this opportunity to raise them now. Thank you for your contributions to this review process.

---

### Author Response · Authors · 2024-11-25
**Code release statement**

Dear AC and reviewers,

We have provided the code through an anonymous link: https://anonymous.4open.science/r/fVLP-DF95 to assist in the review process. We are actively working through our internal approval process and confirm that the code will be released immediately upon approval of our institute.

---

> ### Comment · Area_Chair_iRJK · 2024-11-25
>
> Dear Authors,
>
> Your explanation of the internal approval process is quite clear. However, we would like to understand if there is any considerable possibility that the code release might not be approved. This information will help us better assess the situation during the review process.
>
> Thank you for your clarification.

---

> ### Author Response · Authors · 2024-11-25
> **Code release statement**
>
> It is likely to be approved as we have done before, at least for non-commercial use. Previously we published our models and code on Github and Hugging Face sites.

---

> > ### Comment · Area_Chair_iRJK · 2024-11-26
> >
> > Thank you for the information.

---

### Meta-Review · Area_Chair_iRJK · 2024-12-22

**Metareview:**

This paper introduces a fine-grained vision-language model (fVLM) designed to improve CT image interpretation by aligning specific anatomical regions in CT scans with corresponding descriptions in radiology reports. By addressing the limitations of existing global image-report contrastive methods, the paper proposes a novel anatomy-level contrastive learning framework alongside a strategy to handle false negatives during the learning process. These innovations are particularly impactful in the medical domain, where precise image-text associations are critical. Additionally, the authors present MedVL-CT69K, a large-scale dataset encompassing 272,124 CT scans with expert annotations, which represents a valuable and time-intensive contribution to the field.

The novelty of aligning anatomical regions with descriptive text is well recognised by the reviewers, who unanimously acknowledge its significance. The proposed model achieves promising performance on critical tasks such as zero-shot abnormality detection and report generation. Furthermore, the commitment to releasing MedVL-CT69K strengthens the paper’s impact, as expert annotations in medical imaging are notoriously expensive and laborious to obtain.

Based on the strong novelty, performance, and impact of the contributions, as well as the promise to release the dataset and code, I recommend acceptance with a spotlight presentation. This work is poised to inspire further research and practical applications in the medical AI.

**Additional Comments On Reviewer Discussion:**

During the discussion phase, the authors effectively addressed initial concerns raised by reviewers, demonstrating the robustness of their methodology and results. This thorough engagement, combined with the promising experimental outcomes, has led to unanimous positive recommendations from the reviewers. The proposed dataset and model are expected to enable further advancements in medical image analysis, making the work a significant contribution to the community.

---

### Decision · Program_Chairs · 2025-01-22

Accept (Spotlight)